# Plant AtEH/Pan1 proteins drive autophagosome formation at ER-PM contact sites with actin and endocytic machinery

Pengwei Wang[1,2,6], Roman Pleskot[3,4,6], Jingze Zang[1,2], Joanna Winkler[3,4], Jie Wang[3], Klaas Yperman[3,4], Tong Zhang[2], Kun Wang[2], Jinli Gong[2], Yajie Guan[2], Christine Richardson [1], Patrick Duckney[1], Michael Vandorpe[3,4], Evelien Mylle[3,4], Jindriska Fiserova[1,5], Daniel Van Damme[3,4]* & Patrick J. Hussey[1]*

The Arabidopsis EH proteins (AtEH1/Pan1 and AtEH2/Pan1) are components of the endocytic TPLATE complex (TPC) which is essential for endocytosis. Both proteins are homologues of the yeast ARP2/3 complex activator, Pan1p. Here, we show that these proteins are also involved in actin cytoskeleton regulated autophagy. Both AtEH/Pan1 proteins localise to the plasma membrane and autophagosomes. Upon induction of autophagy, AtEH/Pan1 proteins recruit TPC and AP-2 subunits, clathrin, actin and ARP2/3 proteins to autophagosomes. Increased expression of AtEH/Pan1 proteins boosts autophagosome formation, suggesting independent and redundant pathways for actin-mediated autophagy in plants. Moreover, AtEHs/Pan1-regulated autophagosomes associate with ER-PM contact sites (EPCS) where AtEH1/Pan1 interacts with VAP27-1. Knock-down expression of either AtEH1/Pan1 or VAP27-1 makes plants more susceptible to nutrient depleted conditions, indicating that the autophagy pathway is perturbed. In conclusion, we identify the existence of an autophagy-dependent pathway in plants to degrade endocytic components, starting at the EPCS through the interaction among AtEH/Pan1, actin cytoskeleton and the EPCS resident protein VAP27-1.

[1] Department of Biosciences, Durham University, South road, Durham DH1 3LE, UK. [2] Key Laboratory of Horticultural Plant Biology (MOE), College of Horticulture and Forestry Sciences, Huazhong Agricultural University, Wuhan 430070 Hubei Province, PR China. [3] Department of Plant Biotechnology and Bioinformatics, Ghent University, Technologiepark 71, 9052 Ghent, Belgium. [4] VIB Center for Plant Systems Biology, Technologiepark 71, 9052 Ghent, Belgium. [5] Department of Biology of the Cell Nucleus, Institute of Molecular Genetics CAS, v.v.i., Vídeňská 1083, Prague 14200, Czech Republic. [6] These authors contributed equally: Pengwei Wang, Roman Pleskot. *email: dadam@psb.vib-ugent.be; p.j.hussey@durham.ac.uk

The degradation of damaged or excess cellular components in the vacuole is known as autophagy. This process allows cells to rebalance their energy for plants to develop, or to resist biotic and abiotic stresses[1]. Autophagosomes are double membrane compartments that are formed through vesicle transport and membrane expansion to encapsulate cargo destined for the vacuole for degradation[2]. Autophagy also plays a protective role for the cell by removing stress-induced damaged organelles[3]. Most genes that regulate autophagy are well conserved in plants, animals and yeast, but the molecular mechanism for autophagosome biogenesis is still not clear[1]. For both non-selective (occurring predominantly during nutrient starvation) and selective autophagy, the ER membrane is believed to be the major source of the autophagosomal membrane and this is widely reported in different organisms[2]. However, in plants, the exocytic pathway is also implicated in contributing to the autophagosome membrane and it is hypothesised that the endocytic pathway might also be involved[4]. However, our knowledge is very fragmented and the mechanism is still largely under debate[4–6].

In eukaryotic cells, the actin cytoskeleton is known to interact with membrane compartments through various actin interacting proteins. Proteins that regulate actin polymerization are also found to be associated with the endoplasmic reticulum (ER) and the plasma membrane (PM)[7–9], where they have been found to regulate the biogenesis of autophagosomes: the mechanical forces produced by actin filament assembly are utilized to drive phagophore membrane expansion and the engulfment of autophagy cargo[10–13]. In animal cells, the cytoskeleton-autophagy connection is regulated by a number of actin nucleation-promoting factors (NPF), such as WHAMM, JMY, the WASH complex and the SCAR/WAVE complex, all of which are able to activate the ARP2/3 complex to promote branched actin nucleation[14–16]. However, of the above list, only the SCAR/WAVE complex is present in plants[17]. Plants may therefore employ other NPFs to fulfil their needs for actin-driven autophagy during development and in response to various stress conditions.

In this study, we show that two genes in the Arabidopsis genome, AtEH1 and AtEH2, show homology with Pan1p (therefore, we named them as AtEH1/Pan1 and AtEH2/Pan1). This protein represents a yeast Eps15-like protein which activates the ARP2/3 complex to promote actin polymerization and PM invagination during endocytosis in yeast[18,19]. These AtEH proteins were previously shown to be members of the TPLATE complex (TPC) which regulates plant endocytosis[20–22]. TPC is an ancient protein complex that, along with plants, has so far only been identified in Dictyostelium. In plants, TPC contains eight protein components and the complex is essential for life, knock-out mutants in several of the TPC genes are male sterile. Both AtEH proteins may be auxiliary to the core TPC complex, as in Dictyostelium, these two proteins did not co-purify with the other complex proteins[23].

We show here that AtEH/Pan1 proteins are able to interact with F-actin and also VAP27-1, which is a protein that localizes to the ER-PM contact site[24–26], and regulate the formation of autophagosomes. Activation of ARP2/3 by the SCAR/WAVE complex has been shown to regulate autophagosome formation in plants[11]. Therefore, the data presented in this paper indicate that independent and redundant pathways for actin-mediated autophagy exist in plants and, moreover, that the autophagy pathway can be initiated at the at the ER-PM contact sites (EPCS), a location at which the ER network is physically linked to the PM. Our results here reveal an alternative mechanism for autophagosome formation at plant EPCS[24,27,28], where actin-mediated autophagy converges with the endocytic machinery.

## Results and discussion

**Identification of regulators for actin-dependent autophagy.** The formation of autophagosomes in plants requires both the ARP2/3 complex which nucleates branched actin filaments and the SCAR/WAVE complex which activates the ARP2/3 complex[11,12,15]. Both complexes are essential for autophagy that can be induced by external pressure[11,13]. To better understand this process, we searched for plant homologues of yeast Pan1p as potential activators of the ARP2/3 complex[19]. Two Pan1p homologues were identified in the Arabidopsis genome (Fig. 1a). These have two N-terminal Epsin15-homology (EH) domains, a coiled coil domain and a C-terminal VCA-like domain, an acidic stretch conserved in most ARP2/3 activators[29]. These proteins are also known as AtEH1 and AtEH2 and are both biochemically shown to be part of the octameric endocytic TPLATE complex which is required for plant endocytosis[21]. Therefore, we re-named these proteins AtEH1/Pan1 and AtEH2/Pan1 which is in accordance with our phylogenetic analysis and domain architecture (Fig. 1a). Interestingly, a basal angiosperm plant, Amborella trichocarpa, contains only one EH protein (AtriEH, Fig. 1b), which duplicated independently in monocot and dicots. Our phylogenetic analysis, together with published data showing that yeast Pan1p as well as AtEH1 can homodimerize[30,31], suggests that more primitive TPLATE complexes possibly contain only one EH protein forming a homodimer.

AtEH1/Pan1 and AtEH2/Pan1 are essential for plant life. In AtEH1/Pan1 heterozygous mutants, shrivelled and normal looking pollen grains are observed in a ratio of 1:1 (Fig. 2a), similar to what has been reported for AtEH2/Pan1 and several other mutants in TPC subunits and CME (Clathrin-mediated endocytosis) effectors[21,22]. The failure to transfer the mutation to the next generation via the pollen can be complemented by expressing the respective AtEH/Pan1 proteins fused to mRUBY3 and driven by the Histon3 promotor in the mutant background, indicating that the fusions are functional (Table 1). Both AtEH1/Pan1-mRuby3 and AtEH2/Pan1-mRuby3 localise to the plasma membrane in Arabidopsis roots and hypocotyl cells where they can be visualized localizing to dynamic endocytic foci (Fig. 2d, e). Some discrete cytosolic punctate structures can also be observed with low frequency in root cells (Fig. 2b, c).

To further confirm this result, we performed immunofluorescence studies using an antibody which recognizes both the AtEH/Pan1 proteins. This antiserum identifies two distinct bands on a western blot of a 1D gel of protein extract from Arabidopsis seedlings (Fig. 2f); the upper band at 135 KDa represents AtEH2/Pan1, and the lower band at 110 KDa represents AtEH1/Pan1 based on their respective molecular weights. This antibody is also able to recognize GFP-AtEH1/Pan1 and GFP-AtEH2/Pan1 fusion proteins when over-expressed in N. benthamiana (Fig. 2g). Immunofluorescence analysis confirmed the PM localization of the AtEH/Pan1-mRUBY3 proteins in roots (Fig. 2h–j) and also revealed endogenous AtEH/Pan1 to mark discrete punctate structures in leaf epidermal cells and root hairs. The expression of GFP-AtEH1/Pan1 also revealed punctate structures in Arabidopsis cotyledon and hypocotyl cells (Fig. 2k, l), and the increased presence of autophagosomal structures at the EM level of these plants suggests that the fluorescent puncta are likely to be autophagosome related (Fig. 2m–o).

**Plant AtEH/Pan1 proteins localize to autophagosomes.** In order to characterize the nature of the AtEH1/Pan1-labelled puncta, we used transient expression in N. benthamiana to co-express GFP-AtEH/Pan1 with various markers. In N. benthamiana, GFP-AtEH1/Pan1 and RFP-AtEH2/Pan1 co-localise to puncta (Fig. 3a). A FRET-FLIM study in co-expressing cells

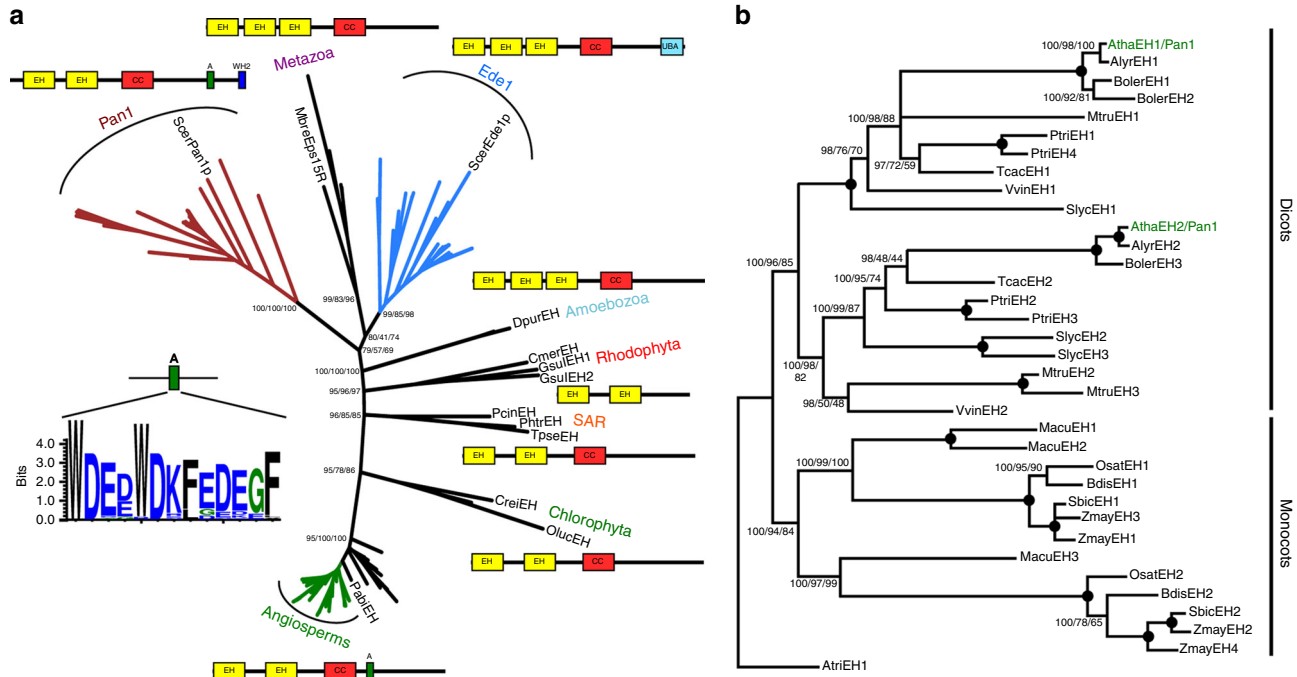

**Fig. 1** Phylogenetic analysis and domain architecture points to both Arabidopsis AtEH proteins as homologues of yeast Pan1p. **a** Phylogenetic tree for Pan1 homologues shows that Plant AtEH/Pan1 proteins form a well-supported clade with a domain organization similar to yeast Pan1p. Interestingly, the Amoebozoa seem to possess homologous proteins with a domain architecture resembling more to Ede1 than to Pan1. A—acidic motif, EH—Epsin15-homology domain, CC - coiled coil domain, UBA—ubiquitin-associated domain, WH2—WASP-homology 2 domain, Dpur—Dictyostelium purpureum, Cmer—Cyanidioschyzon merolae, Gsul—Galderia sulphuraria, Mbre—Monosiga brevicollis, Pabi—Picea Abies, Pcin—Phytophtora cinnamomi, Phtr—Phaedactylum tricornutum, Scer—Saccharomyces cerevisiae, Tpse—Thallasiosira pseudonana. The sequence logo representing the conservation of the acidic motif is shown left of the tree. **b** Detailed tree rooted to Amborella trichocarpa, which only possesses a single EH/Pan1 protein, showing that EH/Pan1 proteins duplicated independently in Dicots and Monocots. Atha—Arabidopsis thaliana, Alyr—Arabidopsis lyrata, Atri—Amborella trichocarpa, Bdis—Brachypodium distachyon, Boler—Brassica oleracea, Mtru—Medicago truncatula, Macu—Musa acuminata, Osat—Oryza sativa, Sbic—Sorghum bicolour, Slyc—Solanum lycopersicum, Vvin—Vitis vinifera, Tcac—Theobroma cacao, Zmay—Zea mays. Numbers at nodes correspond to the posterior probabilities from the Bayesian analysis/RAxML bootstrap support/approximate likelihood ratio test with the SH-like (Shimodaira–Hasegawa-like) support from PhyML

identified a significant drop in the GFP-AtEH1/Pan1 life-time in the presence of RFP-AtEH2/Pan1 (Fig. 3b), indicating that the two proteins interact with each other (likely forming hetero-dimer/oligomers). As they can form at least dimers our further experiments concentrated on one component, AtEH1/Pan1. GFP-AtEH1/Pan1-labelled puncta move along the ER surface, and fuse with each other (Fig. 3c, Supplementary Movie 1). This feature is reminiscent of a previously identified autophagy-actin regulatory protein, NAP1 which is a component of the SCAR/WAVE complex[11]. Co-expression of GFP-AtEH1/Pan1 with the mature autophagosome marker RFP-ATG8e indicates that these structures are indeed autophagosomes (Fig. 3d; Supplementary Fig. 1A). The observation that GFP-AtEH1/Pan1 localises to autophagosomes is further confirmed by immunofluorescence studies where co-localisation of GFP-AtEH1/Pan1 with endogenous ATG8-stained autophagosomes is observed in cotyledons of stable transgenic Arabidopsis lines expressing GFP-AtEH1/Pan1 (Fig. 3e). However, the number of autophagosomes in transgenic Arabidopsis was less than the number observed in transient *N. benthamiana*, this variation likely reflects differences in expression levels of AtEH/Pan1. Interestingly, GFP-AtEH1/Pan1 also co-localizes with the early autophagosome marker, YFP-ATG6 (Fig. 3f), which is known to be recruited to the autophagosomal membrane at the membrane expansion stage prior to the recruitment of ATG8. Similar experiments using C-terminal fusions of AtEH1/Pan1-GFP and AtEH2/Pan1-GFP also co-labelled with ATG8a at punctate structures in *N. benthamiana*

(Supplementary Fig. 1B–C), indicating that the position of the GFP does not affect protein function in autophagy. As a control for the specificity of the autophagosome recruitment, we show that AtEH2/Pan1-mCherry co-expressed with free GFP does not lead to recruitment of the GFP signal to AtEH2/Pan1 (Supplementary Fig. 1D). Partial co-localization can be observed between AtEH1/Pan1-mCherry and NBR1-GFP (an autophagy receptor that binds to ubiquitinated proteins[32], while little co-localization was found with AtEH2/Pan1 (Supplementary Fig. 1E–F). This result could be caused by the fact that AtEH1/Pan1 contains two ubiquitination sites[33], which are not conserved in AtEH2/Pan1. This difference is in agreement with the fact that both AtEH1/Pan1 and AtEH2/Pan1 are not redundant.

Concanamycin A (Conc A) treatment visualized strong accumulation of functional AtEH1/Pan1-mRUBY3 (in the ateh1 (−/−) mutant background) as punctate structures inside the vacuole of Arabidopsis root cells upon carbon stress (Fig. 3g). Moreover, these punctate structures co-localized with ATG8a, confirming their autophagosomal nature (Fig. 3h).

Autophagosomes are enriched in phosphatidylinositol 3-phosphate (PI3P) and proteins that regulate PI3P levels are positive modulators of autophagy initiation[34]. Next, we analysed the effect of Wortmannin (a commonly used PI3K blocker that is known to inhibit autophagy[35,36]) on the localization of AtEH1/Pan1. The number and size of autophagosomes was reduced when *N. benthamiana* leaves co-transformed with GFP-AtEH1/Pan1 and RFP-ATG8e were treated with Wortmannin (Supplementary

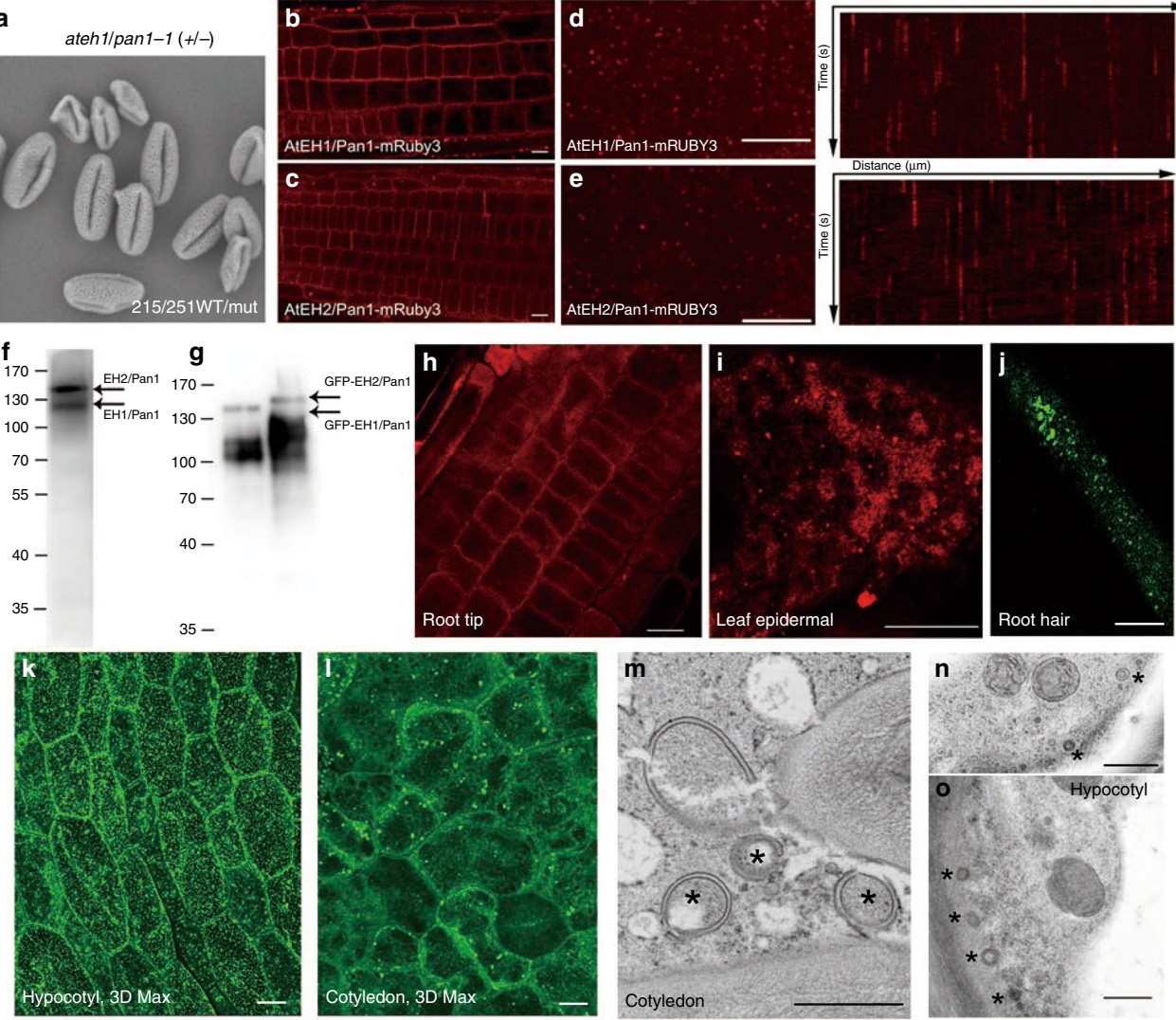

**Fig. 2** AtEH/Pan1 localize to the PM as well as to punctate structures that are likely to be autophagosomes. **a** Representative SEM image of pollen from a heterozygous ateh1/pan1-1 plant showing normal and shrivelled pollen in a 1:1 ratio. **b-e** In the Arabidopsis root (**e**, **f**) and hypocotyl cells (**l**, **m**), AtEH/Pan1-mRuby3 proteins, in their respective complemented mutant background, localize predominantly to the PM at dynamic endocytic foci. To some extent, discrete larger punctae can be observed. **f**, **g** An antibody raised against AtEH2/Pan1 recognizes both proteins in Arabidopsis seedlings (**g**) and recognizes GFP-AtEH/Pan1 fusions transiently expressed in N. benthamiana. **h-j** Immunolocalization using the anti-AtEH/Pan1 antibody confirms the PM localization in Arabidopsis root cells and also identifies punctate labelling in leaf epidermal and root hair cells (Scale bar: 10 μm). **k-o** Punctate structures were identified in cells of Arabidopsis hypocotyl and cotyledons stably transformed with GFP-AtEH1/Pan1 (**k**, **l**). The same samples were further analysed at the ultrastructural level, and numerous double membrane autophagosome-like structures were identified (**m-o**; Scale bar: 500 nm)

Fig. 2A–B), indicating that the labelling of AtEH/Pan1 to autophagosomes is sensitive to PI3P levels.

It is known that cytoplasm to vacuole transport, and therefore also autophagosomal degradation, is blocked in autophagy defective mutants (e.g. *atg5* and *atg7* mutants)[37,38]. In our study, we have stably transformed 35S::GFP-AtEH1/Pan1 in Arabidopsis with different genetic backgrounds (e.g. *Col-0*, *atg5* and *atg7*). We first analysed the localization of GFP-AtEH1/Pan1 in both wild type and in autophagy mutants under non-stressed conditions. No significant difference was observed between either background (Supplementary Fig. 2C–D). For example, AtEH1/Pan1 localized to the PM, as well as the newly formed cell plate, in root meristem cells of wild type and *atg5* Arabidopsis plants, and it still localized to punctate structures, suggesting that its subcellular localization and function in cytokinesis is likely independent of the autophagy machinery.

Subsequently, we analysed the seedlings under stress conditions. Seedlings were first grown in MS medium for 7 days before being transferred to nitrogen deficient medium with or without Conc A. In parallel, Arabidopsis seedlings expressing GFP-ATG8a were used as a positive control to monitor the effectiveness of the treatment. The result clearly demonstrated that in wild type plants, GFP-ATG8a and GFP-AtEH1/Pan1 accumulated inside the vacuole in nitrogen-free medium supplemented with Conc A (Fig. 3i, j). On the other hand, no vacuole internalization of GFP-AtEH1/Pan1 could be identified with the same treatment in *atg5* or *atg7* mutants (Fig. 3k, l). Taken together, our results suggested that the delivery of AtEH1/Pan1-labelled autophagosome to the vacuole requires a functional autophagosomal pathway (Fig. 3i–l); but the formation of the punctae marked by AtEH1/Pan1 is not (Fig. 3i–l). This further supports our conclusion that AtEH/Pan1 proteins are recruited to

**Table 1 Back-cross experiments of ateh1/pan1-1 and ateh2/pan1-1 plants expressing C-terminal fusions with mRuby3 allow transfer of the T-DNA via the male**

| Back cross to Col-0 (♀) | # plants | T-DNA transfer via ♂ |
|---|---|---|
| ateh1/pan1-1 (+/−) ♂ | 12 | 0 |
| ateh1/pan1-1 (−/−) + AtEH1/Pan1-mRuby3 ♂ | 12 | 11 |
| ateh2/pan1-1 (−/−) + AtEH2/Pan1-mRuby3 ♂ | 12 | 11 |

phagophore in the early stages, before ATG8 lipidation and membrane recruitment.

**AtEH/Pan1 proteins interact with F-actin and ARP2/3 subunits**. Yeast Pan1p has been shown to bind to F-actin and activate the ARP2/3 complex[19]. It is therefore likely that AtEH1/Pan1 may regulate actin-dependent autophagy through the ARP2/3 complex in plants similar to that previously described for SCAR/WAVE activation of ARP2/3[17]. Consistent with this hypothesis, a GST-His tagged full-length AtEH1/Pan1 protein was expressed and purified from *E. coli*. The protein was co-incubated with F-actin and subjected to a co-sedimentation assay. Western blotting comparing the supernatant versus pellet fraction in the presence or absence of actin shows that GST-His-AtEH1/Pan1 binds directly to F-actin in vitro (Fig. 4a). In planta experiments also demonstrated that GFP-AtEH1/Pan1-labelled autophagosomes co-align with actin (Fig. 4b), a phenomenon that has been described in previous autophagosome-actin interactions[11,15]. RFP-ARP3, a component of the ARP2/3 complex, co-localises with AtEH1/Pan1 and ATG8-labelled autophagosomes (Fig. 4c). Furthermore, FRET-FLIM revealed a significant drop in the GFP-AtEH1/Pan1 lifetime in the presence of RFP-ARP3, indicating that AtEH1/Pan1 interacts directly with the ARP2/3 complex (Fig. 4d). Taken together, these data indicate direct association of AtEH1/Pan1 with F-actin and ARP2/3, indicating that the machinery for actin remodelling is co-located at autophagosomes.

**AtEHs/Pan1 recruit endocytic proteins to autophagosomes**. AtEH1/Pan1 proteins have previously been shown to be part of the endocytic TPLATE complex (TPC) acting at the plasma membrane to internalize cargo largely in concert with the AP-2 complex[21]. Transient expression in *N. benthamiana*, combining AtEH1/Pan1-mCherry with several TPC subunits showed that, next to TPLATE, other TPC subunits (TML, TWD40-1 and TWD40-2) were also recruited to the AtEH/Pan1-positive autophagosomes (Fig. 4e, f; Supplementary Fig. 3A–F). Triple expression of CFP-ATG8e, TPLATE-GFP and AtEH1/Pan1-mCherry also confirmed the co-localization of these proteins at the autophagosomes (Fig. 4g). To address whether the other subunits of the TPC are involved in the autophagy-related function of AtEH/Pan1, we carbon-starved dual complemented, double ateh1/pan1 (−/−)/tplate (−/−) or ateh2/pan1 (−/−)/tplate (−/−) mutants in the presence of Conc A. Similar to the observed accumulation of AtEH1/Pan1 and ATG8 positive autophagosomes in the vacuole (Fig. 3g, h), we observed co-localization of TPLATE-GFP punctae with both AtEH1/Pan1 and AtEH2/Pan2 positive autophagosomes (Fig. 4h, i). The vacuolar co-localization of TPLATE with AtEH/Pan1 suggests that AtEH/Pan1 proteins, together with other TPC subunits, are likely degraded via the autophagy pathway. In agreement with our microscopical observations, carbon starvation resulted in substantial degradation of TPLATE in comparison to the control

conditions (Fig. 4j). Furthermore, strong vacuolar labelling of AtEH/Pan1-mCherry was specifically found in the red channel after pro-longed expression in *N. benthamiana*, indicating that the vacuole is the destination of the AtEH/Pan1-mCherry-labelled autophagosomes (Supplementary Fig. 3G). The presence and degradation of AtEH/Pan1-mCherry was also confirmed by western blotting after 7 days of infiltration (Supplementary Fig. 3H).

In addition to TPC subunits, AP-2 (AP2A1, AP1/2B1, AP2M and AP2S) subunits and clathrin (CHC1) were also recruited to the AtEH/Pan1-positive autophagosomes (Supplementary Fig. 4; Supplementary Fig. 5). Whereas autophagosomal and endosomal markers localize to distinct structures under normal conditions, strong expression of AtEH1/Pan1 enhanced the co-localization between YFP-ATG8e and the late endosomal marker RabF2a-mCherry (Supplementary Fig. 5B–C), suggesting the AtEH1/Pan1 may facilitate the convergence between the endocytic and autophagy pathways.

Our data indicate that the ARP2/3, TPLATE complex subunits, AP-2 and Clathrin are recruited by AtEH/Pan1 to autophagosomes. The actin cytoskeleton is likely to be required for membrane expansion and movement, whereas the presence of the TPLATE complex, AP-2 and Clathrin may point to endocytosis-derived membrane trafficking events from the plasma membrane contributing membrane and cargo to these autophagosomes. The co-localization of the autophagosomes with late endosomal markers suggests that the AtEH/Pan1-related autophagosomes converge with the endosomal pathway destined for vacuolar degradation. In previous studies, we showed that NAP1 of the SCAR/WAVE complex recruited ARP2/3 and other SCAR complex components onto autophagosome membranes when autophagy was induced[10]. The fact that AtEH/Pan1 can recruit ARP2/3 components to autophagosomes implies that either pathway of actin nucleation is likely to be redundant in their activity.

**AtEH1/Pan1-labelled autophagosomes form at the EPCS**. Autophagosome membranes originate from multiple sources and the ER membrane is believed to be a major compartment for autophagosome formation[2,39]. However, the presence of endocytic machinery at AtEH/Pan1 autophagosomes also implies a contribution from the PM. Our next aim was therefore to determine the identity of the subcellular locations that are required for AtEH/Pan1-mediated autophagosome formation. Immunogold TEM of Arabidopsis root cells revealed that endogenous AtEH/Pan1-labelled autophagosomes were closely adjacent to the ER and PM (Fig. 5a). Similarly, autophagosomes, which are labelled for endogenous NAP1, ATG8 or ATG4, are also found at the ER-PM contact regions (Fig. 5b–d). This result is reminiscent of our previous study where NAP1-labelled autophagosomes partially co-localized with VAP27-1 at the ER-PM contact sites (EPCS)[11]. Therefore, it is likely that autophagosomes can form at the EPCS in plants. This observation is further confirmed in *N. benthamiana* where RFP-ATG8e-labelled autophagosomes are found associated with the VAP27-1-labelled EPCS (Fig. 5e). Our results here indicate that the plant EPCS are able to act as the platform for autophagosome formation by recruiting key regulatory proteins. However, EPCS are unlikely to be the only source for autophagosome formation. For example, the ER-mitochondria contact sites, the ER-Golgi interface, and the entire ER network[2] are also proven to be essential for the formation of autophagosomes, which reflects the diversity of autophagosome origins.

In order to study the possible relationship between EPCS and AtEH1/Pan1 in autophagosome formation, we co-expressed

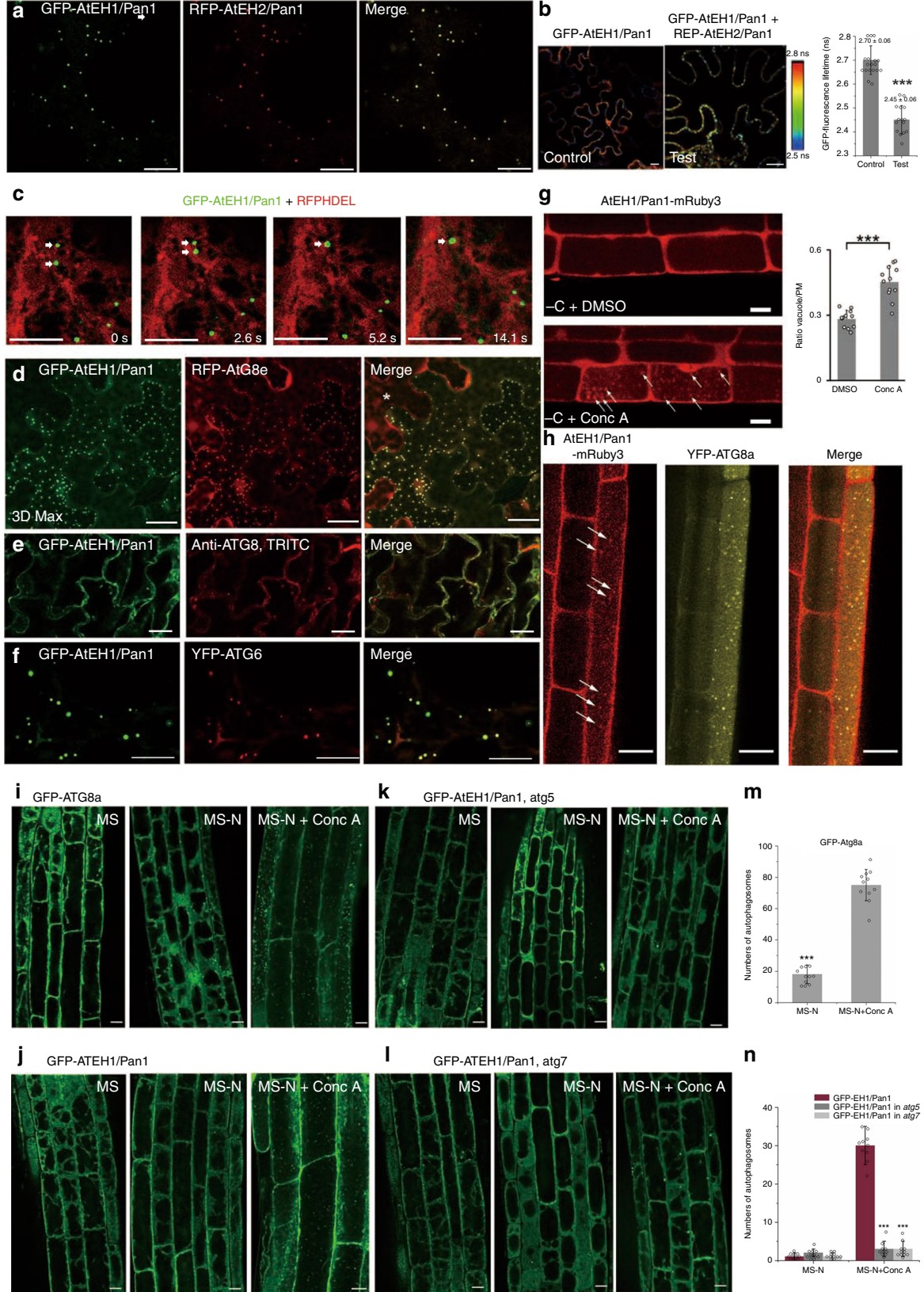

GFP-AtEH1/Pan1 with the EPCS resident proteins, VAP27-1 and SYT1. Transient co-expression of GFP-AtEH1/Pan1 with VAP27-1-YFP in *N. benthamiana* revealed a strong co-localization between AtEH1/Pan1-labelled autophagosomes and VAP27-1-labelled EPCS (Fig. 5f), whereas AtEH/Pan1 localized adjacent to SYT1 (Supplementary Fig. 6A)[40,41], further supporting our conclusion that autophagosomes are formed at EPCS. In animal cells, extended-synaptotagmins (homologues of plant

**Fig. 3** AtEH1/Pan1 localizes to autophagosomes. **a**, **b** GFP-AtEH1/Pan1 and RFP-AtEH2/Pan1 co-localize at punctate structures in *N. benthamiana*, and FRET-FLIM revealed an interaction between them. The fluorescence life-time of GFP-AtEH1/Pan1 (control) changes significantly (half-time reduces from 2.70 ± 0.06 to 2.45 ± 0.06 ns) in the presence of RFP-AtEH1/Pan1 (test condition). **c** When transiently expressed in *N. benthamiana*, GFP-AtEH1/Pan1 localized to mobile puncta that move along the ER network. The puncta fuse homotypically, reminiscent of the behaviour of autophagosomes. **d** Co-expression of GFP-AtEH1/Pan1 with RFP-ATG8e (mature autophagosome marker) in *N. benthamiana* identified almost complete co-localization between the two proteins, indicating that these punctate structures are autophagosomes. Please note the cell lacking GFP-AtEH1/Pan1 expression (marked with an asterisk), where no RFP-ATG8e positive autophagosomal structures are identified (images are 3D projections, Z = 20 μm, 45 slices). **e** Immunofluorescence using anti-ATG8 in cotyledons of Arabidopsis plants over-expressing GFP-AtEH1/Pan1 showing co-localization between GFP-AtEH1/Pan1 punctae and endogenous ATG8. **f** Co-expression of GFP-AtEH1/Pan1 with YFP-ATG6, an early autophagosome marker, identified almost complete co-localization between the two proteins. **g** Arabidopsis transgenic lines expressing AtEH1/Pan1-mRuby3 were carbon starved. Strong vacuolar accumulation of AtEH1/Pan1-labelled punctae (arrows) was found in the presence of Concanamycin A (Conc A) in contrast to the DMSO-treated control. At least three cells of at least three independent plants were imaged and the ratio between vacuolar and PM intensity was quantified (n = 12). **h** Functional AtEH1/Pan1-mRuby3 in ateh1/pan1-1 (−/−) co-localizes with YFP-ATG8 (arrows) inside the vacuole when Arabidopsis plants are carbon starved and treated with Conc A, indicating that AtEH1/Pan1 participates in and is degraded by an autophagy pathway. **i**, **j** Transgenic Arabidopsis plants expressing GFP-ATG8a and GFP-AtEH1/Pan1, grown in different growth media. Autophagosome structures were identified inside the vacuole when plants were grown after nitrogen starvation and treatment with Conc A. **k**, **l** GFP-AtEH1/Pan1 localization in two autophagy deficient mutant backgrounds (atg5 and atg7). Upon nitrogen starvation and Conc A treatment, no punctate structures are found to accumulate inside the vacuole, indicating the transport of GFP-AtEH1/Pan1 punctae into the vacuole relies on a functional autophagy pathway (Scale bar: 10 μm). **m**, **n** Quantification of the results in panels (**i**) to (**l**) (n = 11). Conc A treatment results in significantly higher amount of autophagosomes, marked by GFP-ATG8a (panel **m**) and GFP-AtEH1/Pan1 (panel **n**), inside the vacuole upon nitrogen starvation. In both autophagy mutant backgrounds, GFP-AtEH1/Pan1-positive autophagosomes can still form in the cytoplasm and their amount is significantly reduced compared to the amount of autophagosomes visible inside the vacuole in the WT background. N ≥ 15 for every FRET-FLIM analysis, error bars are S.D., ***P < 0.001 in Student's T-test

SYT1) contribute to the biogenesis of autophagosomes at the EPCS[34]. This suggests that the function of EPCS in regulating autophagy is a conserved process in eukaryotic cells.

GFP-AtEH1/Pan1, VAP27-1-YFP and RFP-ATG8e exhibit a strong co-localization when co-expressed in *N. benthamiana* (Supplementary Fig. 6B). This result is confirmed with immuno-localization studies using anti-AtEH1/Pan1 in the Arabidopsis lines expressing VAP27-1-YFP, and also by the co-localisation of 35S::GFP-AtEH1/Pan1 in the same VAP27-1-YFP Arabidopsis lines. In these experiments the majority of the AtEH1/Pan1-labelled puncta co-localized with VAP27-1-YFP at EPCS (Supplementary Fig. 6C–D). In contrast, anti-AtEH1/Pan1 immunofluorescence using Arabidopsis seedlings expressing GFP-HDEL (an ER marker) showed no co-localization between AtEH1/Pan1 puncta and the ER membrane (Supplementary Fig. 6E), demonstrating the specificity of our observed immunolocalization pattern.

**AtEH1/Pan1 interacts with VAP27-1 at EPCS**. To address whether AtEH1/Pan1 proteins would interact with VAP27, protein-protein interaction studies using FRET-FLIM and GFP-trap assays were performed. The results of these analyses demonstrated that RFP-VAP27-1 and GFP-AtEH1/Pan1 interact in *N. benthamiana* (Fig. 5g, h). The expression of VAP27-1-YFP prevents the movement of AtEH1/Pan1 autophagosomes, most of which are docked at the EPCS and labelled by VAP27-1. As AtEH/Pan1 co-localized with the ATG8-labelled autophagosome in the presence of VAP27-1 (Supplementary Fig. 6B), our result suggests a direct effect of the enhanced interaction between EPCS and the autophagosome membrane through VAP27-1 and AtEH1/Pan1 (Supplementary Fig. 7A–B). GFP-AtEH1/Pan1 also co-localised with other members of the plant VAP27 family and the yeast VAP27 homologue, Scs2 (Supplementary Fig. 8A–D)[42] and requires the VAP27 Major-sperm domain (MSD) (Supplementary Fig. 8E–F, J). Moreover, co-localization and interaction is only observed between VAP27 and the EH1/Pan1 T2 mutant (discussed later), indicating that the two proteins interact through the N-terminus of VAP27 and the C-terminus of AtEH1/Pan1 (Supplementary Fig. 8G–I).

In summary, our results demonstrate the co-localisation of VAP27-1, AtEH/Pan1 and ATG8 in plants. In addition, VAP27-1 also interacts with clathrin in AP-MS [43], which further indicates

the involvement of the endocytic machinery in this process. Previous studies demonstrated that endocytic proteins and VAP27-1 are localized to the cytoplasm/PM or to the ER network under normal conditions, respectively. However, our results demonstrated that these proteins can also be recruited to the autophagosome membranes when autophagy is activated, suggesting that they are also involved in regulating plant autophagy. We therefore proceeded to investigate whether the function of AtEH1/Pan1 and VAP27-1 is essential for the regulation of autophagy in plants.

**The C-terminus of AtEH/Pan1 is essential for autophagy**. To explore the function of different domains of AtEH1/Pan1 in the regulation of autophagosome formation, we truncated AtEH1/Pan1 (T1, aa.1–500, containing the EH domains and T2, aa.474–1019, containing the coiled coil and the acidic motif) and expressed both halves as GFP-fusion proteins in *N. benthamiana* (Supplementary Fig. 9A). Western blot analysis confirmed that these protein truncations are made and stable (Supplementary Fig. 9F). The N-terminal truncation (T1) containing both EH domains predominantly labelled small punctate structures associated with the PM (stained with FM4-64; Supplementary Fig. 9B, D and E), which resemble endocytic foci. The C-terminal part (T2) localized to the cytoplasm (Supplementary Fig. 9C), indicating that the recruitment of AtEH1/Pan1 to membrane requires its N-terminal EH domains.

It is known from literature that the C-terminal acidic domain is responsible for the interaction with ARP2/3 complex and actin cytoskeleton[19]; therefore, we suspect that the AtEH1/Pan1 T1 mutant cannot interact with subunits of the ARP2/3 complex. Indeed, when GFP-AtEH1/Pan1 T1 is co-expressed with RFP-ARP3 and analysed by FRET-FLIM, no interaction could be observed (Supplementary Fig. 9G). In animal and yeast cells, the interaction between AtEH1/Pan1 and ARP2/3 regulates actin polymerization, therefore, we hypothesised that AtEH1/Pan1 T1 mutant without the functional acidic motif may have less autophagy activity because the force generated by actin polymerization to drive membrane deformation is reduced. To test this hypothesis, GFP-AtEH1/Pan1 T1 (lacking the acidic motif) was co-expressed with RFP-ATG8e in *N. benthamiana*. AtEH1/Pan1 T1 labelled punctate structures did not co-localize

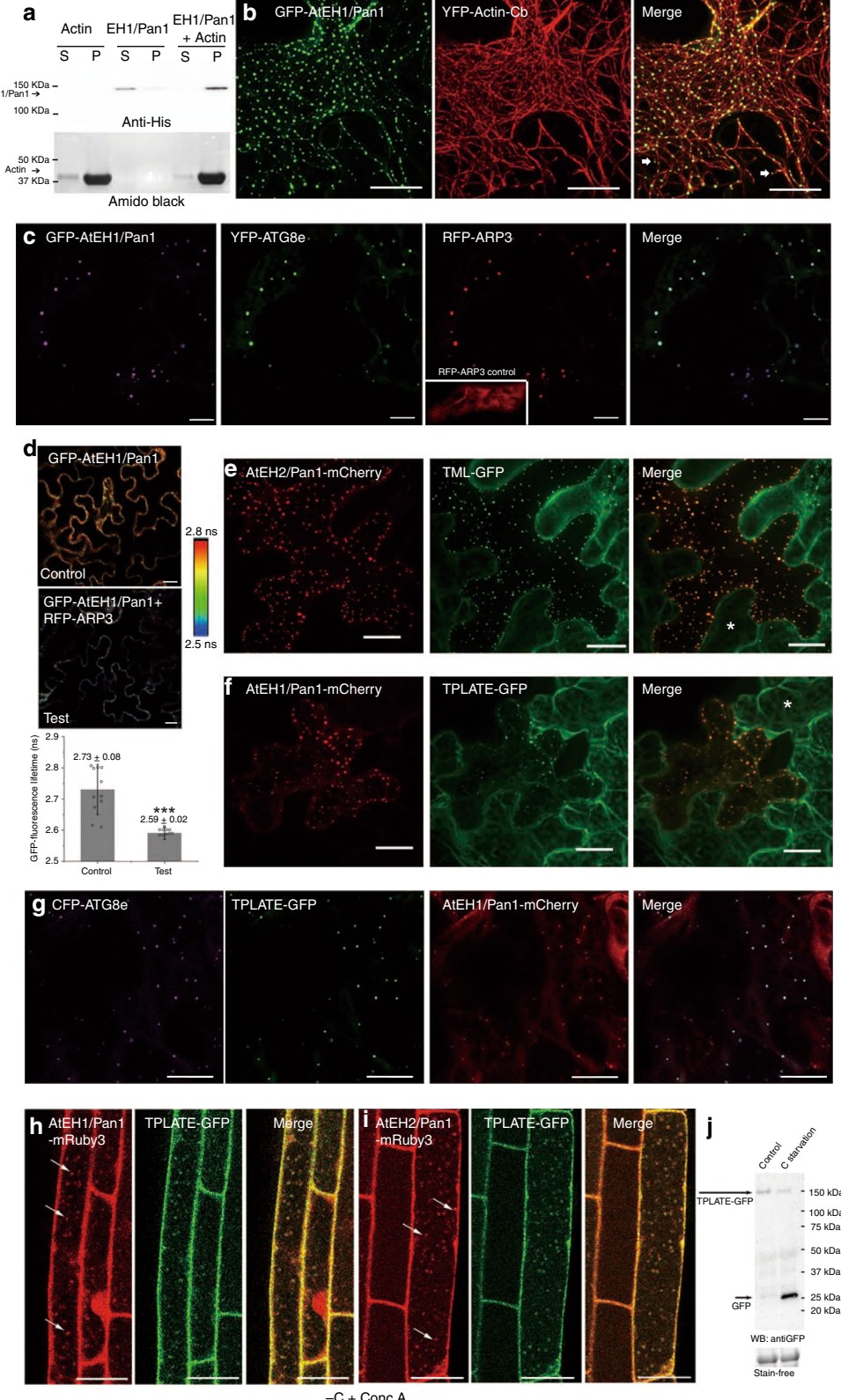

with ATG8, and the number of autophagosomes was much reduced compared to the expression of full-length AtEH1/Pan1 (Supplementary Fig. 9H). Moreover, the expression of autophagy-related genes was not upregulated when the truncated construct was expressed, indicating that the C-terminal part of AtEH1/Pan1 is required to promote autophagy (Supplementary Fig. 9I).

**Overexpression of AtEHs/Pan1 promotes autophagy.** The number of autophagosomes significantly increases with the elevated expression of AtEH1/Pan1 in *N. benthamiana* leaf epidermal cells. ATG genes are also upregulated compared to the vector-only control (Fig. 6a), suggesting that the expression of GFP-AtEH1/Pan1 boosts autophagic activity in these cells.

**Fig. 4** AtEH1/Pan1 binds to the actin cytoskeleton and recruits components of the ARP2/3 and endocytic machinery. **a** Full-length GST-His-AtEH1/Pan1 co-sediments with F-actin *in vitro*, indicating direct actin binding. **b** In *N. benthamiana*, GFP-AtEH1/Pan1-labelled punctae associate at junctions where a few actin filaments intersect, co-align with F-actin forming beads on a string-like pattern or localize to the tip of an actin filament (arrows), thereby confirming the association between GFP-AtEH1/Pan1 and the actin cytoskeleton in vivo (images are 3D projections, $Z = 10\,\mu m$, 25 slices). **c** GFP-AtEH1/Pan1, YFP-ATG8e and RFP-ARP3 co-localize at autophagosomes in *N. benthamiana* leaf epidermal cells. The inset image shows strong cytoplasmic localization of RFP-ARP3 when expressed alone. **d** The fluorescence life-time of GFP-AtEH1/Pan1 (control) changes significantly (half-time reduces from $2.73 \pm 0.08$ to $2.59 \pm 0.02$ ns) in the presence of RFP-ARP3 (test condition), indicating that the two proteins interact (Scale bar: 10 μm). **e, f** TML-GFP and TPLATE-GFP, subunits of the endocytic TPLATE complex, are recruited from the cytoplasm to autophagosomes in the presence of AtEH1/Pan1-mCherry when transiently expressed in *N. benthamiana*. Please note that in the absence of AtEH1/Pan1-GFP expression (cells marked with an asterisk), the localization of TML and TPLATE is cytoplasmic. **g** Triple expression of AtEH1/Pan1-mCherry, CFP-ATG8e and TPLATE-GFP in *N. benthamiana* leaf epidermal cells, showing that the three proteins co-localize at autophagosomes (Scale bar: 20 μm). **h** After carbon starvation and Conc A treatment, AtEH1/Pan1-mRuby3 and TPLATE-GFP, in the complemented ateh1/pan1-1 (−/−)/tplate (−/−) double mutant background, strongly co-localize inside the vacuole (arrows). **i** Similarly, strong co-localization was also observed between TPLATE-GFP punctae and AtEH2/Pan1-mRuby3 positive autophagosomes inside the vacuole after carbon starvation and Conc A treatment. The seedlings are in the ateh2/pan1-1 (−/−)/tplate (−/−)) double complemented, double mutant background (Scale bar: 25 μm). **j** Western blot analysis of functional TPLATE-GFP levels in Arabidopsis tplate (−/−) seedlings with and without carbon starvation showing enhanced degradation of TPLATE during stress conditions. $N \geq 10$ for every FRET-FLIM analysis; error bars are S.D., ***$P < 0.001$ in Student's *t*-test

To check whether increased expression of AtEH/Pan1 would also boost autophagy in Arabidopsis seedlings, we generated β-Estradiol inducible lines for both AtEH1/Pan1 and AtEH2/Pan1, C-terminally fused to GFP. We monitored protein levels and localization of both AtEH1/Pan1-GFP and AtEH2/Pan1-GFP following β-Estradiol induction of gene expression. β-Estradiol-dependent induction of expression was obvious for both constructs starting 12 h post induction and declined at 72 h. At 12 h post induction, AtEH/Pan1 fusions accumulated in the cytoplasm of root meristem cells and the first autophagosomes became apparent in cells in the differentiation zone of the roots (Fig. 6b, c and Supplementary Fig. 10A–B).

At 24 h post induction, AtEH/Pan1 are found at the plasma membrane in the root meristem cells as well as in autophagosomes in differentiation/maturation zone cells (Fig. 6b, c and Supplementary Fig. 10A–B). The shift from cytoplasmic to membrane localization likely reflects the transition from mere overexpression of this protein to the incorporation of AtEH/Pan1 into the TPC[21]. This time point coincides with a rise in ATG8 levels (Fig. 6d and Supplementary Fig. 10C). At 48 h, ATG8 protein levels peak (Fig. 6d and Supplementary Fig. 10C), and this time point coincides with a decrease in GFP signal intensity in the meristem cells, while the cells of the differentiation zone continue to possess multiple autophagosomes (Fig. 6c and Supplementary Fig. 10B). The rise in ATG8 levels following β-Estradiol induction indicates that boosting AtEH/Pan1 expression also boosts autophagy (Fig. 6d; Supplementary Fig. 10C). The observation that the GFP signal disappears earlier in the meristem region compared to the differentiation zone would indicate a higher autophagic flux in these cells compared to differentiated ones. Moreover, mCherry-ATG8e was also transformed into the AtEH1/Pan1-GFP β-Estradiol inducible lines. After induction, strong co-localization between two proteins was found, confirming that also in this model system, these structures are autophagosomes (Fig. 6e). Absence of co-localization between the AtEH/Pan1-labelled autophagosomes and the protein aggregation dye ProteoStat®, excludes the possibility that the increased autophagosome number is triggered by protein aggregation (aggresomes) due to overexpression (Fig. 6f).

**Reducing AtEH1/Pan1 and VAP27 expression blocks autophagy.** If overexpression of AtEH1/Pan1 boosts autophagy (Fig. 6a), then downregulation is expected to do the inverse. We therefore analysed Arabidopsis AtEH1/Pan1 RNAi knock-down lines to further demonstrate the involvement of the respective proteins in autophagy. Autophagy defective mutants (e.g. atg5)

are more susceptible to nutrient deficiency, such as nitrogen starvation (-N), carbon starvation (-C) or low concentration of growth supplements (1/10MS)[38]. To test if AtEH1/Pan1 affects autophagy in plants, nutrient starvation experiments were performed. Independent *AtEH1/Pan1* RNAi transgenic Arabidopsis mutant lines which exhibit roughly 30% knock-down expression were selected for this study (Fig. 6g, h). In these mutants, the expression of selected ATG genes was down-regulated (Fig. 6i), and the number of autophagosomes (labelled by anti-ATG8) was reduced under nutrient depletion conditions (Fig. 6g). Consequently, the *AtEH1/Pan1* RNAi lines were more susceptible to both nitrogen and carbon starvation (Fig. 6h; Supplementary Fig. 10D). Surprisingly, we observed a stronger susceptibility of the RNAi lines to carbon depletion stress than to nitrogen depletion stress, which might reflect stimulus or tissue specificity in autophagosomal pathways[44].

Moreover, we also analysed *VAP27-1* RNAi knock-down and *SYT1* T-DNA knock-out Arabidopsis mutant lines generated in a previous study[25,40]. In contrast to plants grown on normal medium (MS), plant development was significantly delayed in *syt1* and *syt1/VAP27-1* RNAi mutant lines under nutrition deficient conditions (1/10MS). WT Arabidopsis plants did not exhibit a reduced root growth phenotype when subjected to the same treatment (Supplementary Fig. 11A). These results suggest that plant EPCS resident proteins play important roles in response to autophagic stress (nutrition depletion). Although SYT1 does not co-localize with AtEH1/Pan1, previous studies showed that it is required for maintaining the stability of VAP27-1 at the EPCS[40] and it is therefore likely to be indirectly involved in VAP27-1 and AtEH/Pan1 regulated autophagy.

To further support the requirement of EPCS for autophagy, we have looked at the involvement of EPCS during autophagy induction using a stable Arabidopsis line expressing VAP27-1-GFP driven by its endogenous promoter. The number of VAP27-1 labelled EPCS increases when plants are under nitrogen starvation (Supplementary Fig. 11B), suggesting that more EPCS are required when autophagic activity is increased. In yeast and animal cells, EPCS are shown to regulate the local synthesis of phospholipids, such as PI3P, a key regulator of autophagosome formation[34]. Therefore, enhanced ER-PM interaction may be required during autophagy induction, providing additional sites for autophagosome formation[45]. Moreover, a recent study also links SYT1 recruitment to the EPCS under high salt and low nutrient conditions, resulting in expansion of the EPCS[46], similar observations have been reported in animal cells[34].

At this point, we cannot exclude the possibility that the increased susceptibility of *AtEH1/Pan1* and *VAP27-1* RNAi

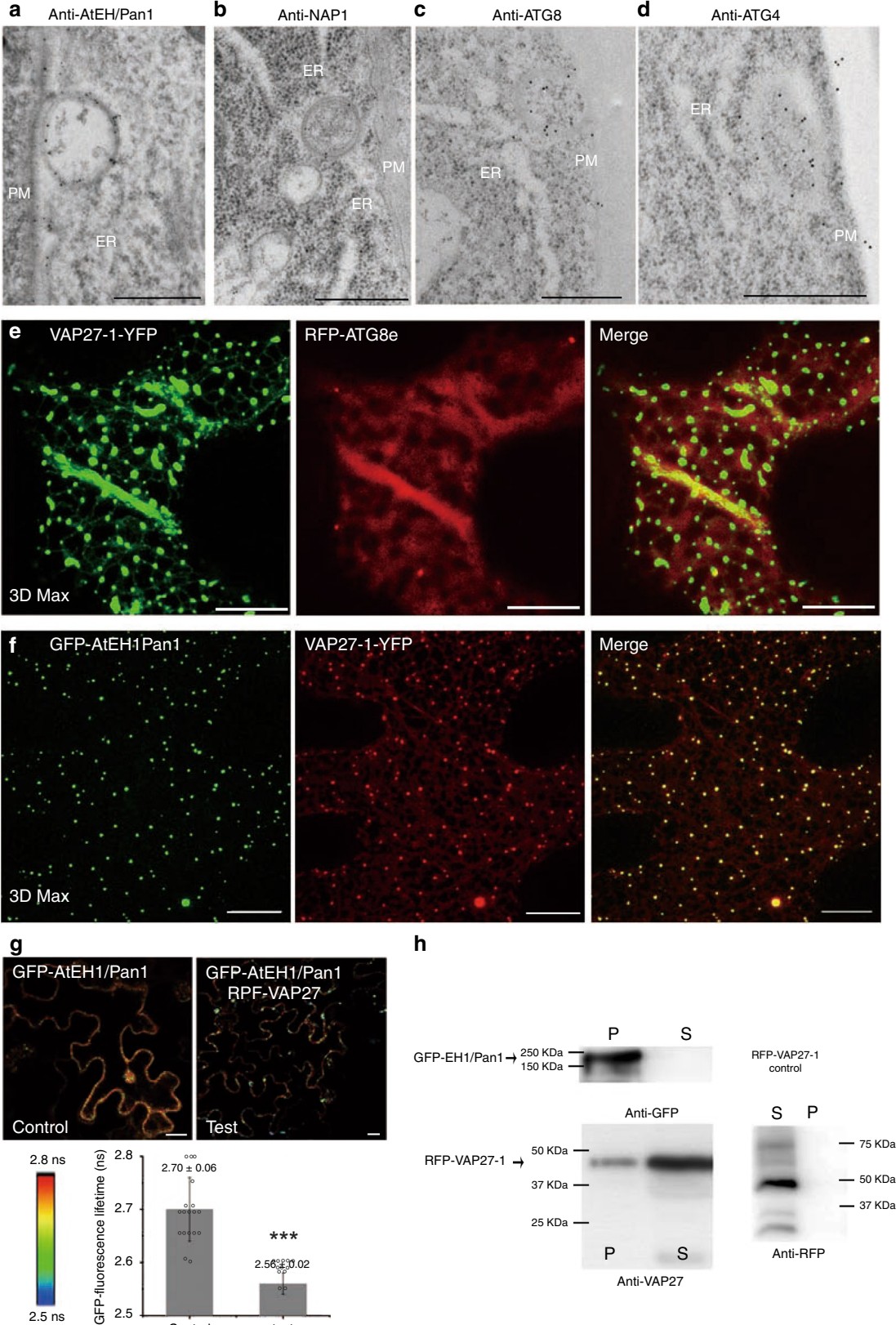

mutants may also reflect disruption of the endocytic pathway. As AtEH/Pan1 proteins, as part of TPC are known to regulate endocytosis[21]. Our data presented here favours the hypothesis that endocytosis at the EPCS is the first step into the autophagic pathway. However, the exact regulation and crosstalk between the endocytotic and autophagy pathways, as well as how the destiny of the endocytosed proteins is regulated requires further investigation.

Our data indicates that AtEH/Pan1 localizes to both inner and outer membrane of the autophagosomes. For example, the

**Fig. 5** AtEH1/Pan1 interacts with VAP27-1 and is involved in the formation of autophagosomes at the ER-PM contact sites. **a–d** Immunogold TEM labelling on Arabidopsis root cells. Autophagosome-like structures that are labelled by anti-AtEH/Pan1, anti-NAP1, anti-ATG8 or anti-ATG4 gold particles are found at the site where the ER and PM are connected (Scale bars: 500 nm). **e** Consistent with this observation, ATG8e-labelled autophagosomes can be found associated with the ER-PM contact sites that are labelled by VAP27-1-YFP. **f** GFP-AtEH1/Pan1-labelled autophagosomes can be recruited to VAP27-1-YFP containing ER-PM contact sites (Scale bar: 10 μm; images are 3D projections, $Z = 18$ μm, 35 slices). **g** Using FRET-FLIM, GFP-AtEH1/Pan1 is shown to interact with RFP-VAP27-1. The fluorescence life-time of the GFP-AtEH1/Pan1 donor protein (control) is on average 2.70 ± 0.02 ns, whereas it significantly reduces to 2.58 ± 0.02 ns in the presence of RFP-VAP27-1 (test condition). **h** The interaction between VAP27-1 and AtEH1/Pan1 is further confirmed using a GFP-trap based co-precipitation assay. RFP-VAP27-1 is found in the pellet fraction in the presence of GFP-AtEH1/Pan1. As control, RFP-VAP27-1 is not able to be precipitated by the GFP-trap. $N \geq 12$ for every FRET-FLIM analysis; error bars are S.D., ***$P < 0.001$ in Student's t-test

interaction between F-actin and ARP2/3 complex suggesting the protein is likely localized to the outer surface (Fig. 4a–c); while its vacuole accumulation after Conc A treatment, together with TPLATE representing other TPC subunits, indicates it is also found inside the autophagosome (Fig. 3g–j). Immunogold labelling of the endogenous AtEH1/Pan1 also identified the existence of gold particles at either side of the membrane (Fig. 5a). Similar localizations have been described for other autophagy regulating proteins, e.g. Exo70B1 and SH3P2[5,47], but their biological relevance remains to be addressed.

In conclusion, we have shown that aberrant expression of AtEH/Pan1 alters the autophagy activity in planta, and that both AtEH1/Pan1 and VAP27-1 mutants are more susceptible in nutrient deficient conditions, indicating that these proteins are essential for the correct execution of autophagy in plants. Based on our results, we propose a model for the formation of autophagosomes at the EPCS in plants, where the actin cytoskeleton, AtEH1/Pan1, ARP2/3 as well as endocytic machinery are involved (Fig. 7). This is a further step forward in our understanding of how autophagy is regulated in plants and the involvement of EPCS in this process.

## Methods

**Phylogenetic analysis**. To identify Pan1p homologues, the predicted proteins of each genome were searched using BLASTP[48] with *Saccharomyces cerevisiae* Pan1p as an input sequence. Used databases were GenBank (https://www.ncbi.nlm.nih.gov/genbank/), Joint Genome Institute (https://genome.jgi.doe.gov/portal/), EnsemblPlants (https://plants.ensembl.org/index.html) and Congenie (http://congenie.org/start). See Supplementary Data 1 for a complete list of all organisms, in total we searched 58 different eukaryotic genomes. Sequences identified as potential homologues were verified by reciprocal blast into the *Saccharomyces cerevisiae* genome. To decipher a domain organization of individual Pan1 homologues, the SMART database was used[49]. Multiple alignments were constructed with mafft algorithms in einsi mode[50] and manually adjusted. Conserved sequence blocks were concatenated using the Jalview program[51] giving the alignment with 381 positions for eukaryotic sequences and the alignment with 843 positions for Angiosperm sequences. Phylogenetic analysis was carried out utilizing RAxML v8.2.9[52], MrBayes v3.2.6[53] and PhyML v3.0[54]. RaxML was run uder the LG model with the γ-model of rate heterogeneity and bootstrapped with 100 pseudoreplicates. PhyML was run under the LG matrix, γ-corrected among-site rate variation with four rate site categories plus a category for invariable sites, all parameters estimated from the data. MrBayes was used to calculate a bayesian tree with the LG amino acid model, where all analyses were performed with four chains and 1,000,000 generations per analysis and trees sampled every 100 generations. All four runs asymptotically approached the same stationarity after first 500,000 generations which were omitted from the final analysis. The remaining trees were used to infer the posterior probabilities for individual clades. The sequence logo of the conserved acidic motif was generated from the multiple alignment by the Weblogo application[55].

**Molecular cloning**. To generate the pDONRP2-P3R-mRuby3 entry clone, the PCR fragment with a stop codon was amplified from pNCS-mRuby3 (Addgene) using the primers mRuby3_attB1_Fwd and Rev (Supplementary Table 1) and cloned into pDONRP2-P3R via Gateway BP reaction (Invitrogen). To yield the complementation construct for ateh1 and ateh2, entry clones of AtEH1/Pan1 and AtEH2/Pan1 without a stop codon[21] were combined with pB7m34GW[56], pDONRP4-P1R-Histone3p[57], and pDONRP2-P3R-mRuby3 in a triple gateway LR reaction (Invitrogen) to generate pH3::AtEH1-mRuby3 and pH3.3::AtEH2-mRuby3, respectively.

In a parallel study, the cDNAs of AtEH1/Pan1 and AtEH2/Pan1 were amplified from Arabidopsis seedling RNA by RT-PCR (Invitrogen) with gene specific primers. The domain deletion mutant of AtEH1/Pan1 T1 and T2 were generated

using specific primers that were able to amplify aa1–500 and aa474–1019, respectively. Primers used for cloning are listed in Supplementary Table 1.

Gateway entry clones in pDONR221 of AP2A1 (At5g22770), AP1/2B1 (At4g11380), AP2M (At4g46630), AP2S (At1g47830), AtEH1/Pan1 (At1g20760) and AtEH2/Pan1 (At1G21630) without stop codons were previously described[21,58]. Entry clones were used in triple Gateway reaction, combining pB7m34GW10, pDONRP4-P1R-35S and pDONRP2-P3R-eGFP or pDONRP2-P3R-mCherry[59] to obtain pB7m34GW-35S::AP2A1-eGFP; pB7m34GW-35S::AP1/2B-eGFP, pB7m34GW-35S::AP2M-eGFP, pB7m34GW-35S::AP2S-eGFP, pB7m34GW-35S::AtEH1/Pan1-mCherry and pB7m34GW-35S::AtEH2/Pan1-mCherry.

Entry clones for CHC1 (Atg11130), TML (At5g57460), TWD40-1 (AT3G50590) and TWD40-2 (At5g24710) were described before[21,60] and were used in a single gateway reaction together with pK7FWG2[56] to yield the 35S-driven C-terminal fusions.

**Arabidopsis mutant characterization and complementation**. SALK_083997 (ateh1/pan1) was obtained from the Nottingham Arabidopsis Stock Centre (NASC) and identified by genotyping PCR using primer combinations (LP-RP and RP-LBaI, Supplementary Table 1).

The transgenic lines were generated by floral dip. Plants heterozygous for the T-DNA insertion in *AtEH1/Pan1* (SALK_083997) and *AtEH2/Pan1* (SALK_0922003) were transformed with pH3::AtEH1-mRuby3 and pH3.3::AtEH2-mRuby3, respectively. Primary transformants (T1) were selected for the complementation constructs on ½ MS plate supplemented with 10 mg/L Basta. T2 plants expressing AtEH1-mRuby3 and AtEH2-mRuby3 were identified by genotyping PCR to identify homozygous lines for the *ateh1/Pan1* and *ateh2/Pan1* insertion mutations, respectively. Genotyping PCR was performed on genomic DNA isolated from rosette leaves. Genotyping primers for *ateh2/Pan1* are described before[21].

For back-cross experiments, the complemented lines of AtEH1/Pan1-mRuby3 and AtEH2/Pan1-mRuby3 as well as the heterozygous mutant plants of ateh1/Pan1 and ateh2/Pan1 were used as male to cross with Col-0 as female. The transfer of the T-DNA was analysed by genotyping PCR.

The complemented lines of AtEH1/Pan1-mRuby3 and AtEH2/Pan1-mRuby3 were crossed with the complemented tplate mutant line *tplate* (−/−) expressing LAT52p::TPLATE-GFP[22]. The F2 plants expressing AtEH1-mRuby3 or AtEH2-mRuby3 combined with TPLATE-GFP in the double homozygous mutant background were identified by genotyping PCR. Genotyping primers for *tplate* (−/−) are described before[22]. Plants expressing pUBQ10::YFP-ATG8a in Col-0[61] were used as male to cross with AtEH1/Pan1-mRuby3 and AtEH2/Pan1-mRuby3 complemented lines, respectively. F1 plants were used for imaging.

**Generation of inducible AtEH1/Pan1 and AtEH2/Pan1 lines**. To generate pRPS5A::XVE:EH1/Pan1-GFP and pRPS5A::XVE:EH2/Pan1-GFP lines, entry clones of AtEH1/Pan1 and AtEH2/Pan1 without a stop codon[21] were combined with pB7m34GW10, pEN-R4-RPS5A-XVE-L1[62], and pDONRP2-P3R-EGFP in a triple gateway LR reaction (Invitrogen).

β-Estradiol induction of the pRPS5A::XVE:AtEH1/Pan1-GFP and pRPS5A::XVE:AtEH2/Pan1-GFP lines was done by transferring 3-day-old seedlings to medium containing β-Estradiol (Sigma-Aldrich) or solvent (DMSO) as a control. β-Estradiol concentration used was 1 μM.

To follow protein levels, western blot analysis was performed. To this end, roots of induced plants and non-induced controls were collected and homogenized with Retch ball mills mixer. Homogenization buffer[63] was added to homogenized root samples and proteins were extracted. Protein levels in the resulting protein extracts (the final supernatants) were measured using the Qubit Protein assay kit (Thermo Fisher). Equal amounts of proteins were subsequently loaded on a protein gel and SDS-PAGE was performed. Proteins were transferred to PVDF membranes (Bio-Rad). For detection, the membranes were incubated in TBST buffer with 5% non-fat milk prior to antibody incubation (anti-GFP HRP conjugated, 1:2,000, Miltenyi Biotec, 130-091-833; rabbit anti-ATG8, 1:500, Abcam, ab77003; anti-RFP, 1:2000, Chromotec 6G6-20) at room temperature for 2 h. After three washes in TBST buffer, the membranes were either immediately developed using an ECL reagent (GE Healthcare, HRP-conjugated primary antibodies) or probed with HRP-conjugated rabbit secondary antibody (1:10,000, GE Healthcare, NA934) or HRP-conjugated mouse secondary antibody (1:10,000, GE Healthcare, NA931) and subsequently developed using an ECL reagent (GE Healthcare). ImageJ software

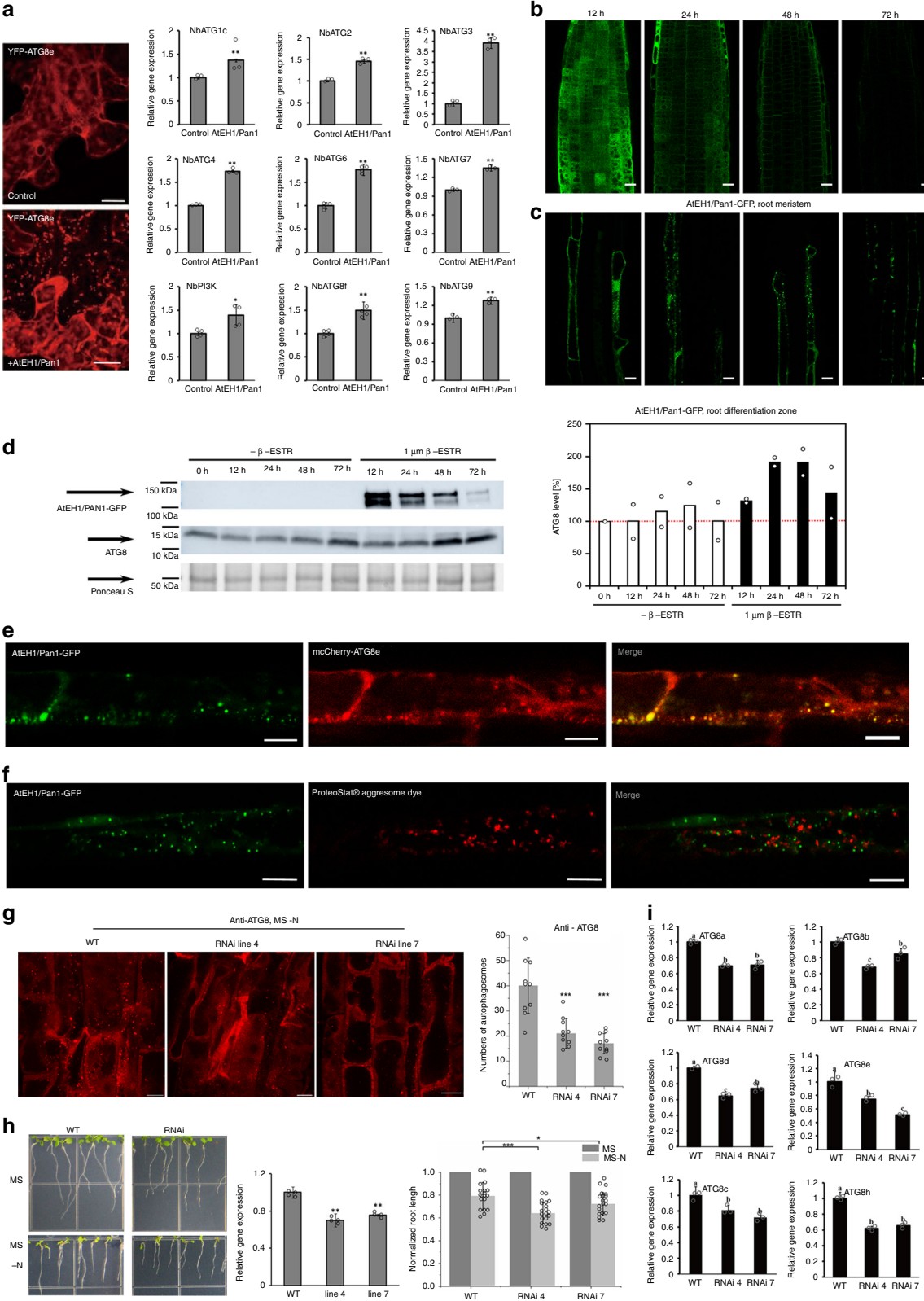

was used to quantify ATG8 levels (https://www.ncbi.nlm.nih.gov/pubmed/22930834).

**Generation of RNAi Arabidopsis plants**. Arabidopsis (Col-0) was grown on either ½ MS agar (1% Sucrose, pH 5.6) or compost in long-day conditions (16 h light, 22 °C and 8 h darkness, 18 °C). Arabidopsis transformations were achieved by floral-dipping. The AtEH1/Pan1 RNAi construct was obtained from AGRIKOLA[64] as an

entry clone, and sub-cloned into the pHELLSGATE RNAi vector[65] which was subsequently used to transform Col-0 Arabidopsis. Positive transgenic plants were selected using appropriate antibiotics and confirmed by RT-qPCR. The *VAP27-1*RNAi Arabidopsis line and the *syt1* mutant used here were generated previously[25,40].

**Autophagy induction with Conc A treatment**. For carbon starvation, Arabidopsis seedlings were grown for 5 days in light followed by 2 days of sucrose starvation.

**Fig. 6** Altered expression of VAP27-1 and AtEH1/Pan1 affects autophagic activity. **a** RT-qPCR of selected autophagy-related genes in *N. benthamiana* leaves transiently expressing either GFP-AtEH1/Pan1 or empty vectors (control). In agreement with the increase in the number of autophagosomes upon AtEH1/Pan1 overexpression, the transcription of most ATG genes is upregulated in the cells transiently transfected with AtEH1/Pan1. The difference between two experiments was calculated by Student's *T*-test, and the significance is indicated with a single ($p \leq 0.05$) or double ($P \leq 0.01$) asterisk. **b, c** Confocal images of root meristems (**b**) and root differentiation/maturation zone cells (**c**) of seedlings taken at the respective time points after transfer to β-Estradiol-containing medium. At 12 h post induction, clear cytoplasmic signal of AtEH1/Pan1 can be detected and the first autophagosomes appear in the differentiation/maturation zone. At 24 h post induction, the signal in the meristem cells is predominantly membrane-associated and autophagosomes in the differentiation/maturation zone are clearly present. At 48–72 h post induction, the signal in the meristem zone decreases, while autophagosomes remain very apparent in the differentiation/maturation zone (Scale bars: 15 μm). **d** After induction by transfer, AtEH1/Pan1 is detected after 12 h and increases up to 48 h. ATG8 protein levels also increase up to 48 h. ATG8 levels of transferred seedlings to non-induced and induced conditions are plotted and show a clear rise from 48 h onward, co-occurring with a decrease in AtEH1/Pan1 abundance. The plot shows the average of two biological repeats. Ponceau S staining was used as loading control. **e** Arabidopsis plants expressing pUBQ10::mCherry-ATG8e and AtEH1/Pan1-GFP, driven by the β-Estradiol inducible promoter. After induction, strong co-localization between the two proteins was found (Scale bars: 10 μm). **f** Absence of co-localization between induced AtEH1/Pan1-GFP punctae and the protein aggregation dye ProteoStat® shows that the AtEH1/Pan1-positive punctae are not aggresomes (Scale bars: 10 μm). **g** Silencing of *AtEH1/Pan1* expression in Arabidopsis (two independent *AtEH1/Pan1* RNAi lines, both of which with about 30% reduction in gene expression) significantly reduced the number of autophagosomes (visualized and measured by the amount of ATG8 positive punctae from an area of 20 × 50 μm, $n = 10$) upon nitrogen starvation, indicating that reduction of AtEH1/Pan1 affects autophagy activity (Scale bars: 10 μm). **h**. *AtEH1/Pan1* RNAi plants exhibit retarded root growth (7.8 ± 1.1 mm and 8.0 ± 1.2 mm, respectively) in nutrient depleted medium (MS -N) compared to the wild type (10.1 ± 1.6 mm). To make each set of experiments comparable, the root length at stress conditions was normalized against its length measured at control conditions ($n = 20$). **i**. Quantitative Real-time PCR of *ATG8* genes in *AtEH1/Pan1* RNAi plants. The expression of most ATG genes is down-regulated in these plants which indicates a reduction in autophagic capacity. Letters indicate statistically different groups based on ANOVA analysis. Error bars are S.D., *$P < 0.05$, **$P < 0.01$, ***$P < 0.001$ in Student's *T*-test

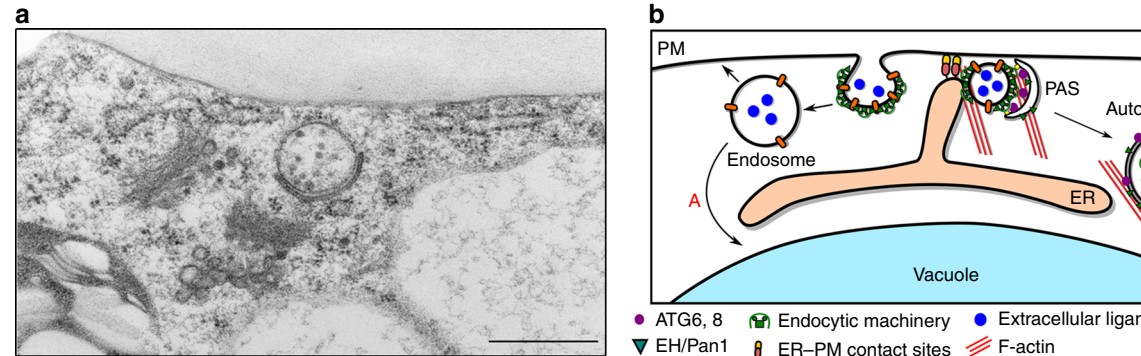

**Fig. 7** Model for AtEH/Pan1 regulated autophagy at the ER-PM contact sites. **a** A representative TEM image showing the formation of autophagosomes at the ER-PM contact sites (Scale bar: 500 nm). **b** Model for autophagosome formation regulated by AtEH/Pan1 and the endocytic machinery at the EPCS. Endocytosis occurs at the ER-PM contact sites through the endocytic machinery that involves the TPLATE complex, AP-2 and clathrin. Endocytosed material can traffic through the conventional pathway, which can either recycle the membrane proteins back to the PM (early endosome) or deliver them to the vacuole (late endosome). In addition, endocytosed material can also be integrated in autophagosomes at the ER-PM contact sites, through the interaction between VAP27-1 and AtEH/Pan1. Internalization by endocytosis is likely shared between the conventional retrograde trafficking and the autophagosomal degradation pathways. Therefore, conditions affecting endocytosis, will also to some extent affect this autophagosomal pathway. Our data are in agreement with a dual function for AtEH/Pan1 proteins. On the one hand, co-localization with ATG6 and interaction with the actin cytoskeleton hints for a role in early autophagosome biogenesis. The presence of AtEH/Pan1 at the PAS (pre-autophagosomal structure) is derived from the co-localization with ATG6. On the other hand, vacuolar delivery and their endocytic role argue for a function in cargo selection for degradation

DMSO or 1 μM Conc A was applied 8–12 h prior to microscopy. For the nitrogen starvation assay, Arabidopsis seedlings were grown in MS medium for 5 days, and transferred onto stress medium (MS -N or MS -N + Conc A) for extra 2 days. For the plant development assays under control and autophagy induction conditions, sterile seeds were grown on vertical plates containing MS, 1/10MS or MS -N/C (without nitrogen or a carbon source) medium for 5–7 days, images were taken and root length was measured using ImageJ.

**Real-time qPCR**. Total RNAs were extracted using Trizol reagent from either *N. benthamiana* leaves or 2-week-old Arabidopsis seedlings according to the user manual (Hipure HP Plant RNA Mini kit, Magen), and single-stranded cDNA was synthesized by a cDNA Synthesis Kit (Ferment). The cDNA of autophagy-related genes (ATG) were amplified with gene specific primers (Supplementary Table 2). For expression analysis, eIF4a and actin1 expression were used as the internal controls for *N. benthamiana* and Arabidopsis, respectively. All qPCR reactions were performed in 384-well plates using a LightCycler® 480 Instrument with four technical replicates. Primer efficiency corrections and melting curve analysis were performed to ensure all reactions produced a uniform and singular product. Results

were evaluated by the 2−ΔΔCp method[66]. All results were derived from three independent experiments with at least three individual plants for each experiment.

**Antigen expression and antibodies production**. A cDNA fragment corresponding to the AtEH2/Pan1 peptide (C-terminal aa1106–1247) which was to be used as an antigen in mice was cloned into the NheI/HindIII sites of the pET28a vector (with N-terminal HisTag) for protein expression. Antigen peptide was expressed in *E. coli* (Rosetta 2, Novagen) by induction with IPTG (1 mM) at 30 °C for 3–6 h. Cells were harvested and resuspended in protein extraction buffer (HEPES, 50 mM pH 7.0; NaCl 300 mM; beta-Mercaptoethanol 5 mM; Urea 8 M). After sonication, the cell mixture was centrifuged at 15,000 rpm and the supernatant was filtered through a 0.4 μm filter to remove cell debris and genomic DNA. The total protein extract was incubated with nickel-agarose beads for 1 h with constant rotating, and the beads were washed three times each with washing buffer 1–3, containing increasing amounts of imidazole (20, 40, 60 mM) and decreasing amounts of urea (6, 4, 3 M). Proteins were eluted with elution buffer (HEPES, 50 mM pH 7.0; NaCl 300 mM; imidazole 250 mM; urea 2 M) and dialysed in PBS

overnight at 4 °C. The purified antigenic protein used for making polyclonal antibodies in mice was described in a previous study[67,68].

**Immunofluorescence**. Immunofluorescence studies on root meristem cells were performed as described in Wang et al. Root tips were fixed in 4% PFA and 0.01% glutaraldehyde, in PIPES buffer containing 0.1 M PIPES pH 6.9, 1 mM MgSO4 and 2 mM EGTA for 60 min. The fixed samples were digested in Driselase (2%) for 7 min to partially remove the cell wall. Samples were then treated with 0.1% Triton for 15 min to permeabilize the membrane, and incubated in primary and secondary antibody for 3 h or overnight at 4 °C. Freeze shattering and immunofluorescence of leaf cells was performed as described before[25,69]. Anti-AtEH/Pan1 and anti-ATG8 antibodies (Abcam, ab98830) were used at 1:100 dilutions, followed by secondary antibody incubation with TRITC or FICT-conjugated secondary antibody (Jackson ImmunoResearch; Code number: 111-025-144; 115-095-166) at 1:200 dilution.

**Electron microscopy and immunogold labelling**. For the TEM study, wild type Arabidopsis (Col-0) were fixed 7 days after germination. Root tip samples were prepared for immunoelectron microscopy by high pressure freezing and freeze substitution. Seeds were germinated on vertically oriented agar plates. After 6–7 days, the distal 1–2 mm tips of the roots were excised with a razorblade, immersed in 20% BSA and quickly loaded into membrane carriers (Leica Micro-systems GmbH) for high pressure freezing with a Leica EMPACT (Leica Micro-systems GmbH). Freeze substitution was performed in a Leica EM AFS freeze substitution device (Leica Microsystems GmbH). The samples were freeze-substituted in anhydrous acetone containing 0.25% (v/v) glutaraldehyde and 0.1% (w/v) uranyl acetate for 48 h at −80 °C then slowly the temperature was raised to −50 °C over a 30 h period. After several rinses in anhydrous acetone at −50 °C, the samples were teased from the membrane carriers with a fine needle. Infiltration continued at −50 °C into Monostep Lowicryl HM20 (Agar Scientific) by increasing the concentration of resin to acetone, 12 h in 22, 33 and 66% and then 96 h in 100% (three changes). Final embedding and UV polymerisation was carried out at −50 °C for 48 h followed by a slow warming to 20 °C, the polymerisation then continued for a further 24 h. Ultrathin sections, 50–70 nm, were cut on a diamond knife (Diatome, USA) and collected onto formvar-coated nickel grids[70].

For immunogold labelling sections were blocked for 5 min at room temperature with 1% (w/v) BSA in phosphate buffered saline. The sections were then incubated for a further 30 min with the primary antibody in 0.1% (w/v) BSA in phosphate buffered saline. For ATG4 and ATG8 labelling, antibodies[71] were used at a 1:50–1:100 dilution and detected using 5 nm gold-conjugated goat anti-mouse antibody. For EH/Pan1 labelling, antibody was used at 1:10 dilution. After washing in PBS (3 × 5 min), the sections were incubated with Alexa Fluor® 647 - FluoroNanogold™ Fab' goat anti-mouse IgG (Nanoprobes, Cat number: 7502) diluted to 1:100 for 30 min. For negative controls the primary antibody was omitted. After washing in 1% (w/v) BSA in phosphate buffered saline (3 × 5 min) followed by Milli-Q water (10 × 1 min) the nanogold was then enhanced with GoldEnhance™ EM Plus (Nanoprobes, USA) (1 × 3 min). For TEM observation the sections were stained with 1% (w/v) aqueous uranyl acetate for 5 min, followed by Reynolds lead citrate for 5 min. Sections were examined with a Hitachi H-7600 TEM operating at 100 kV fitted with an AMT Orca-ER digital camera (Advanced Microscopy Techniques, Danvers, USA).

**Scanning electron microscopy (SEM)**. SEM performed on ateh1 heterozygous mutant pollen was done using a Hitachi table top microscope TM-1000 described as previously[21].

**GFP-trapping and western blotting**. Pull-down assays were performed using GFP-Trap_A beads (Chromotek). Approximately 0.1–0.2 g of N. benthamiana leaf material expressing test proteins was ground in liquid nitrogen and resuspended in lysis buffer containing 10 mM Tris-HCl, pH 7.5, 150 mM NaCl, 0.5 mM EDTA, 1 mM phenylmethylsulfonyl fluoride, 0.5% Triton-X100 and Complete Protease Inhibitor (Sigma). The mixture was incubated on ice for 30 min and then cen-trifuged at 10,000 × g for 10 min at 4 °C. The supernatant was transferred into fresh tubes without any cell debris. The GFP-Trap_A beads were equilibrated in 500 ml of dilution buffer containing 150 mM NaCl, 10 mM Tris-HCl, pH 7.5, and 0.5 mM EDTA, and added to the plant extract. The mixture was incubated on ice with constant shaking for 2 h, then tubes were centrifuged at 2500 × g for 2 min at 4 °C, the supernatant and agarose pellet was harvested in separate tubes for analysis. For detection using western blot, protein extracts were heated to 95 °C in SDS buffer for 5 min. Samples were loaded onto a 12.5% SDS-gel followed by electrophoresis and protein transfer to nitrocellulose membrane. For detection, the membrane was incubated in 2xTBST buffer with 5% milk prior to primary antibody incubation (1:200 for anti-Pan; 1:500 for anti-GFP, Abcam ab290) at room temperature for 3 h. After three washes in TBST buffer, the membrane was probed with HRP-conjugated rabbit secondary antibody (BBI Life Sciences, D110058) at 1:3000 and developed using an ECL reagent (GE Healthcare).

**Actin co-sedimentation assay**. The experiment was performed according to Deeks et al.[72] with minor modifications. The full-length AtEH1/Pan1 cDNA was cloned into the SacI/XhoI sites of the pET42a vector (with N-terminal GST and

HisTag) for protein expression. The construct was transformed into E. coli strain BL21 (Novagen), and protein expression was induced by IPTG (1 mM) at 160 overnight. GST-His-Tagged fusion protein was purified using Ni-NTA resin (Qiagen) according to manufacturer's instructions. For co-sedimentation assays, proteins were dialysed in reaction buffer (4 mM Tris pH 8.0, 0.2 mM DTT, 0.4 mM ATP, 20 mM KCl, 4 mM imidazole, 2 mM EGTA, 0.4 mM MgSO4) overnight. Rabbit muscle actin (Cytoskeleton Inc.) suspended in G-buffer (2 mM Tris pH 8.0, 0.5 mM DTT, 0.2 mM CaCl2, 0.2 mM ATP) was polymerized by the addition of 10× KME (500 mM KCl, 10 mM MgSO4, 10 mM EGTA, 100 mM imidazole, pH 6.5) and incubated at room temperature for 2 h. Meanwhile, AtEH1/Pan1 protein solution was centrifuged at 500,000 × g for 30 min to remove any insoluble fractions that may interfere with the experiments. Then, F-actin at a concentration of 5 μM, was mixed with proteins in reaction buffer. The mixture was incubated at room temperature for 15 min and centrifuged at 350,000 × g for 20 min at room temperature. The supernatant and pellet fractions were mixed with SDS buffer and analysed by western blotting using an anti-Histag (Biorbyt, orb344412) antibody.

**Fluorescence probe staining**. For plasma membrane staining, samples were incubated in solution containing FM4-64 (Invitrogen) at 5 μM for 5–10 min before imaging. To stain aggregated proteins, plants were incubated with the ProteoStat aggresome detection reagent (Enzo) in liquid ½ MS medium without sucrose (2 μl of the dye/ml) for 30 min prior to confocal imaging. To visualize autophagosomes, the roots of Arabidopsis seedlings were submerged in solution containing 50 μM MDC (monodansylcadaverine, Sigma) for 60 min prior to confocal imaging.

**Tobacco infiltration, live cell imaging and FRET-FLIM**. Nicotiana benthamiana plants were grown in a growth room or greenhouse with long-day conditions. Transient expression was performed by leaf infiltration according to Sparkes et al.[73]. Transiently transformed N. benthamiana were imaged two days after infiltration using a laser scanning confocal microscope (LSCM, Leica SP5, Leica SP8X or Olympus FluoView FV1000). For each experiment, at least three inde-pendent infiltrations were performed. Images were taken in multi-track mode with line switching when multifluorescence was used. For GFP/RFP or FITC/TRITC combination, samples were excited at 488 and 543 nm and detected at 510–550 and 590–650 nm, respectively. For GFP/YFP combination, GFP was excited at 458 nm and detected at 470–510 nm; YFP was excited at 514 nm and detected at 550–580 nm.

Imaging of AtEH/Pan1, ATG8 and TPLATE in Arabidopsis seedlings was performed on a Leica SP8X microscope equipped with a white light laser. Images were taken using a 40x water objective (40x HC APO CS2, NA = 1.10). To image the GFP signal, the white light laser was set up to 488 nm excitation and a hybrid detector (HyDTM) was used to detect emission between 495 and 550 nm with gating 0.3–6.0 ns. To image the YFP signal, the white light laser was set up to 514 nm excitation and the hybrid detector was used to detect emission between 520 and 550 nm with gating 0.6–6 ns. The mRuby3 was excited by the white light laser at 558 nm and detected with the hybrid detector between 565 and 650 nm with gating 0.6–6.0 ns. Scan speed was 400 or 600 Hz at a resolution of 1024 × 1024 pixels.

To capture the dynamics of AtEH/Pan1 at the plasma membrane, seedlings were imaged using a Nikon Ti microscope equipped with an Ultraview spinning-disk system (PerkinElmer) and a 512x512 Hamamatsu ImagEM C9100-13 EMccd camera. Images were acquired with a 100x oil immersion objective (NA = 1.45). Hypocotyl cells of 4-day-old etiolated seedlings were imaged with 561 nm excitation light, an emission window between 570 and 625 nm and an exposure time of 500 ms/frame. The images were acquired at a fixed rate of 1 s per time point for 2 min. The kymographs were generated by the Volocity software package (PerkinElmer).

Imaging the co-localization of the endocytic players (TPC, AP-2, clathrin) at autophagosomes together with AtEH/Pan1 in N. benthamiana was done on the Ultraview spinning-disc system using sequential imaging with a 60x water immersion objective (NA = 1.20). 488 nm laser excitation combined with a single band pass filter for GFP (500–550 nm) and 561 nm laser excitation combined with a dual band pass filter (500–530 nm and 570–625 nm) for RFP. Z-stacks were acquired with a 1 μm interval using the Ultraview (piezo) focus drive module. Images shown are Z-stack projections except the images showing vacuolar accumulation, which represent a single slice (Fig. S3F and G).

Fluorescence of AtEH1/Pan1-GFP or AtEH2/Pan1-GFP co-infiltrated with 3xGA-TagRFP-AtAtg8a was analysed with an inverted confocal microscope FluoView FV1000 (Olympus), equipped with a 60x water-corrected objective (NA 1.2). Imaging was performed in a multichannel setting with 488 and 559-nm excitation light for GFP and TagRFP excitation, respectively. Fluorescence was captured in the line-scanning mode using an emission window between 500 and 531 nm and between 570- and 630-nm band-pass emission windows for GFP and TagRFP fluorescence, respectively. Images shown represent a single slice (Fig. S1).

FRET-FLIM experiments were carried out using a LeicaSP5 LSCM installed with the fluorescence lifetime system (PicoQuant). The GFP-AtEH1/Pan1 alone was used as donor whose fluorescence lifetime was measured as the negative control. All measurements were taken from whole field images expressing fluorescence protein at similar levels, the average and standard error of different fluorescent lifetime were calculated from at least 15 independent measurements, and the significance of the result was analysed by Student's T-test.

Constructs or FP markers used in this study are listed in Supplementary Table 3.

**Accession numbers**. The Arabidopsis Genome Initiative locus identifiers for the genes mentioned in this article are AtEH1/Pan1 (AT1G20760), AtEH2/Pan1 (AT1G21630), ARP3 (AT1G13180), RabF2a (At5g45130), ATG8e (AT2G45170), ATG8a (AT4G21980), ATG6 (AT3G61710), TPLATE (AT3G01780), TML (At5g57460), TWD40-1 (At3g50590), TWD40-2 (At5g24710), AP2A1 (At5g22770), AP1/2B1 (At4g11380), AP2M (At4g46630), AP2S (At1g47830), CHC1 (AT3G11130), VAP27-1 (At3g60600), VAP27-3 (At2g45140), VAP27-4 (At5g47180), VAP27-8 (At4g21450), SYT1 (AT2G20990) and NBR1(AT4G24690).

**Reporting summary**. Further information on research design is available in the Nature Research Reporting Summary linked to this article.

## Data availability

The authors declare that all data supporting the findings of this study are available within the article and its Supplementary Information files, or from the corresponding author upon reasonable request. Source Data underlying Figs. 3, 4, 5, 6 and Supplementary Figs. 2, 8, 9, 10, 11 are included as a Source Data file.

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

## Acknowledgements
The authors would like to thank Julie Merchie (Ghent University/VIB-PSB) for help with the initial transient *N. benthamiana* experiments. Prof. Savvas Savvides (Ghent University/VIB-IRC) for constructive discussion and Professor S.P. Dinesh-Kumar and Ugrappa Nagalakshmi from the Department of Plant Biology, The Genome Center, University of California, Davis, CA 95616 for sharing their ATG8 marker constructs. The work was supported by NSFC grants (91854102; 31772281), Fundamental Research Funds for the Central Universities (2662018PY010) and Thousand Youth Talents Plan Project to P.W., and a BBSRC grant (BB/G006334/1) to P.J.H. The research in the D.V.D. lab is funded by the European Research Council T-Rex project number 682436 and by the National Science Foundation Flanders (FWO; G009415N).

## Author contributions
P.W. and R.P. performed most of the experiments and wrote the paper with D.V.D. and P.J.H.; D.V.D and P.J.H. supervised the research; J.Z. performed the FRET-FLIM experiments; J.G. and Y.G. contributed to real-time PCR analysis; K.W., J.F., T.Z. and P.D. helped with making AtEH1/Pan1 constructs, generating Arabidopsis transgenic lines and confocal microscopy. C.R. performed the HPF and TEM. J.Wa. characterized the AtEH/Pan1 mutants and the functional fusions. J.Wi. and E.M. performed *N. benthamiana* co-expression analysis of AtEH/Pan1 and endocytic proteins. M.V. and K.Y. generated and helped to characterize the β-Estradiol inducible AtEH/Pan1 lines.

## Competing interests
The authors declare no competing interests.
