## [Peer Review File · Nature Communications]

Reviewers' comments:

Reviewer #1 (Remarks to the Author):

Wang et al.

In this manuscript Wang and Pleskot et al., studied a potential connection between the endocytic machinery and autophagy. Their main claims are (i) T-plate subunit Pan1 drives autophagosome formation (ii) at ER-PM contact sites (iii) together with actin cytoskeleton and endocytic machinery. They also suggest Pan1 can recruit components of the T-plate complex to autophagosomes and to the vacuole. And finally, claim that they have identified a pathway that degrades endocytic components. Although the idea and some of the preliminary data are very exciting, I have several concerns with the manuscript. I hope they help the authors in improving their manuscript?

(1) They have provided extensive evidence showing colocalization between Pan1 and ATG8. There is indeed a massive boost of autophagosomes when Pan1 is coexpressed with ATG8. Some of these experiments are done in a transient system, which is fine. But to make sure that Pan1 is not aggregating and inducing autophagy that way, they should (i) express native promoter driven Pan1 in Arabidopsis GFP-ATG8 lines or at least perform immunofluorescence of Pan1 in GFP-ATG8 expressing lines (ii) co-express Pan1 with NBR1 (aggrephagy receptor) to make sure that Pan1 autophagosomes are not aggregates but autophagosome containing endocytic components.

(2) Figure 3G and H, the signal of the Pan1 in the vacuole seem rather weak. These experiments should be presented as a +/- conA +/-starvation. We should see Pan1 colocalization increasing upon carbon starvation and the number of colocalized foci should increase further upon conA addition. Additionally, the microscopy should be quantified and further supported by WBs (similar set up, +/- C starvation, +/- conA)

(3) Figure 3 J-I: it is very difficult to conclude from the images presented that some of the foci are PM localized. The authors should present these images as colocalization with a PM marker. Additionally, they should show westerns of these truncations to make sure that stability of T2 is similar to T1.

(4) As Pan1 binds to actin, presumably it binds to the outer membrane of the autophagosomes. How can it end up in the vacuole if it is binding to the outer membrane?

(5) FRET-FLIM experiments are great but they would become more significant if the authors included some controls. For example, can they use other actin nucleating factors or Pan1 truncations.

(6) The images presented in 3L should be complemented by westerns. The puncta are rather weak. Again the experiments should be presented as +/- starvation, +/-conA.

(7) Figure S3: how can the authors conclude that the Pan1 puncta are autophagosomes in this experiment specifically? ATG8 is not labeled. Also if they want to make a causal link, they should perform these experiments in an autophagy mutant background or autophagy silenced background. In addition conclusion of B-E is contradicting with H-I. In B the authors claim both TWD40-1 and TWD40-2 are recruited by Pan1 to the autophagosomes, which I think is questionable as I explained above; then in H-I they claim specificity?

(8) 7 days post infiltration seem like a very long time for agroinfiltration experiments. The vacuolar signal could be the autofluorescence of a dying cell. They should show the expression of the

protein with a western 7 days after infiltration.

(9) I think one of the key experiments that are missing at the moment is the analyses of Pan1 recruiting T-plate complex to autophagosomes in an atg mutant such as atg5 or 7 etc. What happens to Pan1 localization? Are T-plate subunits still recruited to the autophagosomes? This would also be a nice complement to wortmannin experiments as wortmannin affects a lot of things and may not “clearly show recruitment of autophagosomes depend on lipid binding”.

(10) In Figure 5 the electron micrographs only show a small portion of the cell. The authors should present less zoomed images where we could see most of a cell and appreciate the accumulation of autophagosomes at ER-PM contact sites. As Pan1 seem to really boost autophagosomes, it should be obvious in un-zoomed images. Also in ATG4 micrograph, I don't see autophagosome-like structures. It may be worth considering to replace this micrograph. In Vap27 micrograph the gold particles seem to be equally present in non-autophagosomal parts as well. Also, the PM is not in proximity of the autophagosome. Is it because of the plane of sectioning? Again for presentation purpose it may be worth replacing this micrograph with one that is similar to the micrograph presented in A.

(11) In figure 6, the authors show RNAi lines of Pan1 is more sensitive to nutrient starvation and conclude that Pan1 is important for autophagy. How can they make a causal link as Pan1 also plays a role in endocytosis. If they silence other endocytic machinery genes, would they not see this sensitivity? Similarly, Syt1 and Vap27 experiments have the same issues. They need to have other controls or cross their RNAi lines or mutants with autophagy mutants to check for genetic interactions. Otherwise, I could imaging affecting endocytosis would have an impact on nutrient starvation sensitivity.

Minor comments:

- Line 30: are found associated with could be replaced by associates with
- No need to tell this to these authors but shorter sentences are easier for readers to get their message. For example 1st sentence of the introduction. Also, it may be worth writing separate results and discussion sections. In the middle of the results section, they have done a recap of their results, which is unorthodox.
- Line 43: should be animals
- Line 45: nutritional should be nutrient
- Line 46: in a variety of different organisms could be in different organisms
- Line 61-70: this is not introduction, it belongs to the results section.
- Line 80: Results and discussion
- Line 95: references not formatted properly
- Line 99: CME has not been described before
- Figure 3K-EH domain seems to prefer PIP2 rather than PI3P. Don't see why the authors focus on PI3P.

Reviewer #2 (Remarks to the Author):

This study attempts to link TPC components (AtEH1 and AtEH2) with the regulation of autophagosome formation at the ER-PM contact sites in Arabidopsis. Unfortunately the authors were unable to provide mechanistic role of AtEH1 and AtEH2 in the regulation of autophagy. Despite large amount of protein localization data, key experiments are missing, as described in the major comments.

Major comments

1. Fig. 2B. T-DNA inheritance experiments lack statistical analysis.
2. Fig 3B, 4D and 5H. Control cells show strong signal in cytoplasm and nucleus, contradicting expected localization of AtEH1/Pan1. It is not clear if fluorescence in the nuclei was included into FRET-FLIM quantification and if so, what is the biological relevance of this data. Are FRET-FLIM results statistically significant? The decrease in the fluorescence lifetime is around 5%. The charts shown with trimmed Y-axis can be misleading. What do error bars show?
3. Fig. 3E and F. Authors do not elaborate on the differences in the number of structures, localization of GFP-AtEH1/Pan1 and colocalization efficacy.
4. Lines 220-224. This is too stretching; It reads like ER-PM contacts are exclusive sites for AtEH-labeled autophagosome. In vivo live imaging analysis using TIRF microscopy is required to conclude EPCS origin of AtEH-labeled autophagosomes.
5. Authors should provide analysis of AtEH/Pan1 localizations in the ATG-deficient background and elaborate on the lack of endocytic impairment phenotypes in the atg knockout lines of Arabidopsis thaliana.
6. Lines 267-270, Fig. 6A. Increased number of YFP-ATG8 foci and elevated expression of ATG genes do not tell whether autophagic flux is enhanced. The authors should run Western blot YFP cleavage assay in the presence/absence of Concanamycin A. An alternative could be NBR1 Western blot. The same problem with Figure S9.
7. Lines 275-293. Increased accumulation of non-lipidated ATG8 protein and AtEH-GFP foci does not indicate enhanced autophagic activity. What about ATG8 lipidation? Is autophagic flux also increased?
8. Fig. 6B-D. What does chart on panel B show? Statistical analysis is missing. There is no difference in the starvation response between WT and RNAi. ATG knockout mutants should be used as control. Autophagic flux should be measured instead of ATG8 gene expression, which is not directly indicative of autophagic activity.
9. Lines 305-313. Again, these experiments do not tell about autophagy level.
10. Fig. S11A. Statistical treatment is missing.
11. Generally it is not clear from the presented data, whether proposed upregulation of autophagy in AtEH/Pan1 overexpressors and susceptibility of AtEH/Pan1 KD to starvation is caused by pleiotropic effect on endocytosis or by autophagy-specific response.

Minor comments

1. Fig. 1. VCA-like domain is not shown.
2. Fig 2A. Pollen viability is typically assessed using Alexander staining. SEM is a suboptimal approach as shrinkage might be caused by exposure to vacuum.
3. Immunofluorescent images lack negative controls
4. Fig. 3B. Scale bars are missing
5. Fig 4A lacks the input and has a weird negative actin-like band for AtEH1/Pan1 sample
6. Lines 182-184. This conclusion is far-fetched. Have the authors indeed shown that AtEH proteins are required for autophagosome formation? Rephrase
7. Lines 192-193. How can vacuolar co-localization shown in Fig. 3L suggest functional role of TPLATE in autophagy? Rephrase
8. Lines 210-212. Perplexing, rephrase. Reference(s) is missing
9. Lines 212-214. Premature conclusion. The authors have not shown yet that AtEH proteins are required for autophagosome formation.
10. Fig 5D, there are no autophagosomes on the micrograph
11. Fig 5G contradicts the statement that VAP27-1 is localized at EPCS.
12. Fig. 3D, 5E and F. GFP-AtEH1/Pan1 shows 100% co-localization with RFP-ATG8e and VAP27-1-YFP positive puncta, while only a few RFP-ATG8e co-localize with VAP27-1-YFP puncta.
13. Sections starting on lines 215 and 228 can be combined.
14. Line 229. "AtEH1 regulated autophagy". When readers have reached this point they do not know yet that AtEH1 regulates autophagy.
15. Lines 249-251. This conclusion would be much stronger if also/instead shown for GFP-ATG8 labeled autophagosomes.
16. Please revise the abbreviations used in the text, some lack explanation, e.g. CME.

Reviewer #3 (Remarks to the Author):

This paper describes a comprehensive set of experiments implicating the Arabidopsis proteins, AtEH1 and AtEH2, in actin-mediated autophagy. The authors show that these EH domain-containing proteins, which have previously been implicated in endocytosis as part of the TPLATE complex, link the process of autophagy with endocytosis and the actin cytoskeleton. A diverse set of experiments were conducted to support the authors conclusions including AtEH1 and AtEH2 downregulation, cell biological studies to demonstrate autophagosome localization of functional AtEH1/2-fluorescent protein fusions as well as colocalization of AtEH1 and AtEH2 with markers to various components of the TPLATE complex and ARP2/3, in vitro actin binding, and immunogold electron microscopy. Overall, this is a nice and thorough study. The manuscript is well written and provides a timely follow up on previous work by the authors implicating the actin nucleation

machinery in mediating autophagy in plants. In previous work, they showed that a component of ARP2/3, NAP1, is involved in actin-mediated autophagy. Their discovery that AtEH1 and AtEH2 are part of this actin-modulated machinery is novel and exciting. However, I list some concerns below that the authors might want to address:

1). The paper contains a significant amount of data showing colocalization between AtEH1/2-GFPs and various markers such as autophagosomes, TPLATE components and ARP3. Looking at the image data alone, it is clear that AtEH1/2 colocalizes with these various components. That said, there appears to be a glaring lack of quantitative data to support their colocalization work. Would a Pearson's correlation analysis be a useful quantitative tool for this? I ask for some sort of quantitative data to support the colocalization images because in looking at the image data alone, it appears that in some cases, colocalization is almost 100% (e.g. GFP-ATEH1 and RFP-ATG8e in Figure 3D). Is it always the case that colocalization is almost 1:1 between AtEH-GFP and other red-emitting RFP markers? I would have raised a concern about bleedthrough but I note from the methods section that the authors take this into account by simultaneous scanning. Nonetheless, some sort of quantitation of colocalization would improve the paper and allow more transparency as it would provide information on how many cells were actually sampled, how many independent infiltration/imaging experiments were conducted etc. Such information should be stated in the figure legends.

2). In relation to point 1 above, the statistics used to support data in which authors have done some sort of quantification, need clarification. For example, the FLIM dataset is nice. However, I think the authors missed indicating what type of statistics was done on the FLIM work. I assume "test" in the parenthesis in the figure legend is Student t-test? What are p values? Put asterisks on the FLIM bar graphs to indicate significance. Same point I made above, the authors should indicate in the FLIM analysis, how many cells were sampled. I know they state 15 cells in the methods section but to be more transparent, this information should be indicated the figure legend.

3). As a follow up to my two earlier concerns, the authors might want to check carefully for any missing information in their figure legends for data (and supplemental figures) in which quantification/statistical analysis is involved. Another example are results on low nutrient sensitivity of AtEH/PAN1 knockdowns (Figure 6B). What is the Y axis for Figure 6B. Root length? It seems like the RNAi lines depicted in the bar graphs were not subjected to statistical analysis. Such analysis is needed to show that knockdowns are indeed more sensitive than wild type to nutrient (low N and low C) stress.

4). The authors show that AtEH1 and AtEH2 proteins associate with foci through the N-terminal EH domains. Based on lipid overlay assays, the authors show that such association might be due to binding of AtEHs to anionic phospholipids, most notably PI3P. This dataset could be strengthened by another set of controls or if not too difficult verified by other phospholipid binding assays that is more robust than lipid overlays. With regard to the additional controls, the lipid overlays shown in Figure 3K only have lipid blots for the His-tagged EH domains (i.e. HIS-EH1.1 and HIS-EH1.2). No negative controls are provided. It might be important to include HIS-CC-A (i.e. HIS c-terminus) or HIS alone as negative controls. I raise this issue because these lipid overlay blots can be notorious for non-specific binding. The negative controls would assure readers that the binding patterns shown in Figure 3K are genuinely related to the EH domains. Better yet, the authors might want to consider liposome assays as an alternative to lipid overlay blots.

5). Is the RFP-ARP3 fusion used by the authors to demonstrate colocalization with AtEH1/2-GFP and autophagosome marker functional (Figure 4C)? I think this would be useful for the readers to know. arp3 mutants have a distinct trichome phenotypes so expression of the RFP-ARP3 in the arp3 mutant should complement the trichome defects if it is functional. I might just be missing a reference or prior work by the authors demonstrating functionality of the ARP3-RFP thus this concern could be addressed by simply adding the appropriate reference. I ask about functionality

of the ARP3-RFP because it would potentially be a useful tool for many readers interested in plant actin.

6). The authors demonstrate that overexpressing AtEH/PAN1 promotes autophagy. 24h postinduction, they show that AtEH/PAN1 is at the plasma membrane in the root meristem and are more prominent in autophagosome foci in the elongation zone. I have reservations about the latter claim (Figure 6F). The images of root cells shown with foci clearly have emerging root hairs. Thus, this region of the root can no longer be called the elongation zone because root hairs typically emerge in the differentiation/maturation zone. If the authors want to claim that AtEH/PAN1 is more prominent in the elongation zone compared to the meristem, they need to show cells in the elongation zone that have not yet reached full maturity. I think a few microns or mm from the regions shown in Figure 4E should have cells in the "true" elongation zone. The authors can then claim that AtEH1/PAN1 are in foci in both the root elongation and differential zone.

7). The references in 15 and 16 (Coutts et al) is a duplicate

Reviewer #4 (Remarks to the Author):

This study provides direct evidence showing that AtEH1 and AtEH2 are able to interact with F-actin and ER-PM contact site resident protein VAP27-1, and regulate the formation of autophagosome. Since both AtEH1 and AtEH2 were members of the TPLATE complex (TPC) which regulates plant endocytosis, another new finding in this manuscript is that AtEH1 protein could recruit both TPC and ARP2/3 protein to the autophagosome, indicating the alternative mechanism for autophagosome formation at plant EPCS. In plant cell, the knowledge of how actin participates in autophagosome formation is still very limited. So the current study is potentially of great interest to the field. The experiments are in general well designed and performed, but there were a few major concerns to be addressed for the authors:

1. The authors reported AtEH1 protein drive autophagosome formation at the ER-PM contact site. For this point: immuno-gold TEM of AtEH1 data was provided, but it is difficult to specify the EPCS localization of AtEH1 protein from the single images in Figure 5A, the corresponding statistic analysis should be involved to confirm the AtEH1's specific localization at EPCS or at least similar distribution pattern as VAP27-1. Also the localization of VAP27-1 is different when co-expressed with ATG8 (Fig 5E) compared to when co-expressed with AtEH1 (Fig 5F) in *N. Benthamiana*, is this due to the specific interaction between AtEH1 and VAP27-1 to mediate autophagosome formation? To figure out this point: the co-localization relation between AtEH1, ATG8 and VAP27-1, especially in *Arabidopsis* should be analysed.

2. The authors showed the evidence that AtEH1 co-localized with both ATG8 and ATG6, further the co-localization with ATG6 indicated AtEH1's function in early stage of autophagosome formation, also in the Fig 7B, the AtEH1 were labelled in the PAS. Please provide the direct evidence to confirm this result, for example the time course dynamic distribution pattern of both ATG8 and AtEH1.

3. Different truncated construct were made and confirmed that the EH domain is essential for the lipid binding (Fig 3K), whether the construct with EH domain alone could label punctate structures and PM associated foci as AtEH1-T1 which contain two EH domain? In the end of the study, it is showed the AtEH1-T1 poorly co-localized with ATG8, could authors clarify what the punctate structure is and whether this structure related to the autophagosome formation?

4. To figure out AtEH1's important role in autophagosome formation, the RNAi lines were constructed and some autophagosome phenotypic analysis were made. Would be very important

and interesting to check in these RNAi lines, how ATG8-RFP affected, and also whether the autophagosome formation, VAP27-1 and ARP protein localization will be affected?

In addition, there are some minor concerns are listed:

1. Line 95: Sanchez-Rodriguez et al., Mol Plant 2018, not cited in the proper way, also not involved in the reference list;
2. In the reference list: #15 and #16 are exactly the same.

Reviewers' comments:

Reviewer #1 (Remarks to the Author):

Wang et al.

In this manuscript Wang and Pleskot et al., studied a potential connection between the endocytic machinery and autophagy. Their main claims are (i) T-plate subunit Pan1 drives autophagosome formation (ii) at ER-PM contact sites (iii) together with actin cytoskeleton and endocytic machinery. They also suggest Pan1 can recruit components of the T-plate complex to autophagosomes and to the vacuole. And finally, claim that they have identified a pathway that degrades endocytic components. Although the idea and some of the preliminary data are very exciting, I have several concerns with the manuscript. I hope they help the authors in improving their manuscript?

(1) They have provided extensive evidence showing colocalization between Pan1 and ATG8. There is indeed a massive boost of autophagosomes when Pan1 is coexpressed with ATG8. Some of these experiments are done in a transient system, which is fine. But to make sure that Pan1 is not aggregating and inducing autophagy that way, they should (i) express native promoter driven Pan1 in Arabidopsis GFP-ATG8 lines or at least perform immunofluorescence of Pan1 in GFP-ATG8 expressing lines (ii) co-express Pan1 with NBR1 (aggrephagy receptor) to make sure that Pan1 autophagosomes are not aggregates but autophagosome containing endocytic components.

We are happy that the reviewer finds our data exciting.

We believe the reviewer wants us to rule out that the co-localization we observe between AtEH/Pan1 and ATG8 is caused by over-expression induced aggregation of AtEH/Pan1.

In the original version of the manuscript, we included a pH3.3-driven ATEH/Pan1-mRuby3 fusion which fully complements the mutant's male sterility phenotype. In those weak expression lines, we mostly do not observe aggregation in roots, and AtEH1/Pan1-ATG8 co-localization can clearly be observed when autophagy is induced (Figure 3H). We have adapted the text to make this clearer. The immunolocalization data we provide in Figure 2J and Figure 5A shows that AtEH/Pan1-punctae form in wild type leaves in the absence of stress and that we can localize AtEH/Pan1 to the rim of double-membrane autophagosome-like structures.

To better establish the functional link between AtEH/Pan1 and autophagy, we now performed immunolocalization of ATG8 as well as carbon starvation experiments (in addition to N starvation) in the AtEH/Pan1 RNAi lines (new panel Figure 6G). Furthermore, we transformed GFP-AtEH1/Pan1 into the *atg5* and *atg7* mutant backgrounds and analyzed autophagy flux (new panels in Figure 3). We observe a strong reduction of ATG8-positive punctae upon autophagy induction when AtEH/Pan1 levels are reduced (new panel Figure 6G). Furthermore, the AtEH/Pan1 RNAi lines are similarly hypersensitive to nutrient starvation as *atg5* mutants (new panels Figure S10). We also observed reduced accumulation of GFP-AtEH1/Pan1 in the vacuole under -N + Conc A conditions in both *atg5* and *atg7* mutant backgrounds (new panels Figure 3I-L).

The results above allow us to conclude that AtEH/Pan1 plays an important role in autophagosome formation.

To address the comment regarding aggregate formation upon over expression, we performed additional experiments in *N. benthamiana* as well as in our inducible Arabidopsis lines. When we Combined NBR1 with AtEH1/Pan1 and AtEH2/Pan1 in *N. benthamiana*, we could observe

differential behavior of both AtEH/Pan1 isoforms with respect to NBR1 colocalization. Whereas AtEH1/Pan1 displayed partial colocalization with NBR1, this was not the case for the AtEH2/Pan1 punctae (new panels Figure S1E-F). We believe this differential behaviour confirms the non-redundancy of both AtEH/Pan1 proteins and that the observed colocalization with NBR1 could be linked to the observed ubiquitination of AtEH1 at two sites (Liu et al., 2018 Plant Science) which are not conserved in AtEH2. To investigate this further, we combined our inducible AtEH1/Pan1 OE lines with the mCherry-ATG8e marker and showed strong colocalization between both proteins (new panels Figure 6E). Furthermore, we applied the proteostat aggregation marker to our induced AtEH1/Pan1 lines. We did not observe colocalization between the AtEH1/Pan1 positive autophagosomes and the aggregation marker, showing that overexpression of AtEH1/Pan1 does not result in aggregation in Arabidopsis (new panels Figure 6F). We could not perform this experiment in *N. benthamiana* due to cross-talk of the dye with Chlorophyll autofluorescence. We believe that our current experiments, together with the Wortmannin data in *benthamiana* (Figure S2B) where the autophagosomes disappear upon pharmacological inhibition of PI3P production strongly indicate that the AtEH1/Pan1 autophagosomes are not mere aggregates. We have discussed these findings in the text.

(2) Figure 3G and H, the signal of the Pan1 in the vacuole seem rather weak. These experiments should be presented as a +/- conA +/-starvation. We should see Pan1 colocalization increasing upon carbon starvation and the number of colocalized foci should increase further upon conA addition. Additionally, the microscopy should be quantified and further supported by WBs (similar set up, +/- C starvation, +/- conA)

We would like to point out that the signal is weak because we are working with complemented mutant lines with low expression levels of AtEH1/Pan1 (pHistone H3 vs. 35S for ATG8)

Under normal growth conditions, AtEH1/Pan1 labelled puncta are not abundant (Figure 2 C,D,I; Figure 3J and 3K-L). To show that the accumulation in the vacuole upon nutrient stress is coming from AtEH1/Pan1-mRuby3 (new panel Figure 3G), we repeated the starvation experiment in the presence and absence of Concanamycin A on our complemented AtEH1/Pan1-mRUBY3 lines and we quantified vacuolar accumulation in at least 3 different cells in three independent plants. Our results show a clear and statistical significant difference in vacuolar accumulation upon Conc A treatment. We respectfully disagree with the reviewer on the additional value of confirming this result by WB.

(3) Figure 3 J-I: it is very difficult to conclude from the images presented that some of the foci are PM localized. The authors should present these images as colocalization with a PM marker. Additionally, they should show westerns of these truncations to make sure that stability of T2 is similar to T1.

We performed co-localization of T1 constructs with FM4-64 as PM dye and we performed WB analysis to show stability of these constructs. We added new panels to Figure S9, showing the colocalization of the T1 foci with the FM4-64 dye, the WB result showing that the proteins are made and stable as well as a Z-stack series to clearly demonstrate that the foci are located near the PM.

(4) As Pan1 binds to actin, presumably it binds to the outer membrane of the autophagosomes. How can it end up in the vacuole if it is binding to the outer membrane?

Our data indeed hint to the fact that AtEH/Pan1 binds both inner and outer membrane, just like ATG8, exo70B1 and SH3P2 (Zhuang et al., plant cell 2013; Zhuang and Jiang, autophagy 2014; Kulich et al 2013).

Our pull-down data together with colocalization data show indeed that AtEH1/Pan1 binds actin and that therefore it should be on the outside of the autophagosome. This is also in agreement with our IEM data (Figure 5A).

Furthermore, we have *N. benthamiana* and *Arabidopsis* data showing that AtEH1/Pan1 can also end up in the vacuole, indicating it should be inside the autophagosome as well. We discuss this in more detail in our revised version of the manuscript.

(5) FRET-FLIM experiments are great but they would become more significant if the authors included some controls. For example, can they use other actin nucleating factors or Pan1 truncations.

We agree with the reviewer and we added an experiment where we combined AtEH1/Pan1 T1 with RFP-VAP27 and RFP-ARP3, which shows no interaction as control for the FRET-FLIM. We added these novel data to Figure S8 and S9. We believe using other actin nucleating factors or Pan1 truncations are not ideal as we do not know in advance whether other actin nucleators might interact with AtEH/Pan1 and how AtEH proteins might interact with each other. We have unpublished data from crosslinking proteomics that AtEH proteins can interact head-to-tail. Truncated AtEH/Pan1 constructs are therefore not by default good negative controls.

(6) The images presented in 3L should be complemented by westerns. The puncta are rather weak. Again the experiments should be presented as +/- starvation, +/-conA.

We believe this question is similar to question number 2. In the current Figure 4, we work with double complemented mutant lines (tplate and ateh/pan1). That is why the signal is weak as compared to lines over expressing ATG8.

We believe that the quantification we performed in Figure 3G shows that the vacuolar accumulation is a result of AtEH/Pan1 not being degraded in the presence of Conc A. We have adapted the figure and now provide clear examples of vacuolar colocalization between both AtEH/Pan1 proteins and TPLATE. Furthermore, we also included a novel WB experiment where we compare TPLATE levels in *Arabidopsis* seedlings with and without nutrient stress showing enhanced degradation of TPLATE during stress conditions. We believe these novel findings corroborate our imaging data.

(7) Figure S3: how can the authors conclude that the Pan1 puncta are autophagosomes in this experiment specifically? ATG8 is not labeled. Also if they want to make a causal link, they should perform these experiments in an autophagy mutant background or autophagy silenced background. In addition conclusion of B-E is contradicting with H-I. In B the authors claim both TWD40-1 and TWD40-2 are recruited by Pan1 to the autophagosomes, which I think is questionable as I explained above; then in H-I they claim specificity?

We show in Figure 3, Figure 4 and Figure S1 that the punctae formed by AtEH/Pan1 co-localize very convincingly with ATG8. We think the presence of the ATG8 marker sufficiently defines these punctae as autophagosomes.

To confirm that the colocalization between AtEH1/Pan1 and the other markers in Figure S3 represents autophagosomes, we performed a triple transformation in *N. benthamiana*, combining AtEH1/Pan1-mCherry with TPLATE-GFP and CFP-ATG8. All three proteins show clear colocalization. We added these novel data to Figure 4 panel G.

We believe that our data do show that there is specificity in recruitment. In panels B and E of Figure S3, we show that AtEH1/Pan1 can recruit both TWD40-1 and TWD40-2. In panels C and F, we show that AtEH2/Pan1 recruits TWD40-1 but does not recruit TWD40-2. We believe that this

differential recruitment reflects differences in interactions between both AtEH/Pan1 isoforms. We are only over expressing two subunits here, so more direct interactions will be more visible than those requiring N benthamiana bridging proteins which are present at endogenous levels.

We adapted the figure and included arrows to make it clearer that AtEH1/Pan1 can recruit TWD40-2, while AtEH2/Pan1 does not.

With respect to the causal link between AtEH/Pan1 and autophagosome formation, we refer to our answer on the reviewer's first comment.

(8) 7 days post infiltration seem like a very long time for agroinfiltration experiments. The vacuolar signal could be the autofluorescence of a dying cell. They should show the expression of the protein with a western 7 days after infiltration.

We respectfully disagree with the reviewer that the fluorescence observed originates from a dying cell. To further confirm this, we performed WB analysis and also imaged the cells in different channels as autofluorescence will be visible with BFP and GFP settings as well. Our novel data shows that there is still mCherry present 7days after infiltration and we could clearly observe stronger signal in the red channel compared to the other ones, ruling out that the fluorescence detected originates from autofluorescence of a dying cell. We added these data to Figure S3 panel G and H.

(9) I think one of the key experiments that are missing at the moment is the analyses of Pan1 recruiting T-plate complex to autophagosomes in an atg mutant such as atg5 or 7 etc. What happens to Pan1 localization? Are T-plate subunits still recruited to the autophagosomes? This would also be a nice complement to wortmannin experiments as wortmannin affects a lot of things and may not "clearly show recruitment of autophagosomes depend on lipid binding".

We believe the reviewer asks what happens to AtEH/Pan1 and, in extension, to other TPLATE subunits in case autophagy is defective. We have tried to answer this by combining GFP-AtEH1/Pan1 with atg5 and atg7 mutants. We now included data showing that in those mutant backgrounds, AtEH/Pan1 is still recruited to the PM and cell plate (new panel in Figure S2C) and can still be recruited to punctae which are not delivered to the vacuole (Figure 3K-L). We would like to stress out that we have shown before that AtEH/Pan1 proteins robustly co-purify with all other TPC subunits (Gadeyne et al., 2014). Also here, we provide evidence that AtEH/Pan1 can recruit other subunits in Arabidopsis (Figure 4H and 4I) and in N. benthamiana (Figure 4E-G). We would also like to point out that the lines we work with in Figure 4 (TPLATE-GFP, AtEH/Pan1-mRUBY3) are double complemented (male sterile) mutants. Combing these lines with an atg5/7 mutant background therefore would require generating higher order mutants, which in our view would strongly delay publication without providing a significant addition to the current story.

(10) In Figure 5 the electron micrographs only show a small portion of the cell. The authors should present less zoomed images where we could see most of a cell and appreciate the accumulation of autophagosomes at ER-PM contact sites. As Pan1 seem to really boost autophagosomes, it should be obvious in un-zoomed images. Also in ATG4 micrograph, I don't see autophagosome-like structures. It may be worth considering to replace this micrograph. In Vap27 micrograph the gold particles seem to be equally present in non-autophagosomal parts as well. Also, the PM is not in proximity of the autophagosome. Is it because of the plane of sectioning? Again for presentation purpose it may be worth replacing this micrograph with one that is similar to the micrograph presented in A.

We would like to point out that the images in figure 5A-D come from non-stressed plants and we therefore do not claim autophagy is boosted under these conditions. Non-zoomed images therefore will not reveal any massive accumulation of autophagosomes.

We believe that the fact that Pan1 boosts autophagosomes is already apparent from Figure 2N-O.

We provided a better image for ATG4 (Figure 5 panel D). For simplicity, and also because another reviewer had an issue with this, we removed the IEM micrograph on VAP27.

(11) In figure 6, the authors show RNAi lines of Pan1 is more sensitive to nutrient starvation and conclude that Pan1 is important for autophagy. How can they make a causal link as Pan1 also plays a role in endocytosis. If they silence other endocytic machinery genes, would they not see this sensitivity? Similarly, Syt1 and Vap27 experiments have the same issues. They need to have other controls or cross their RNAi lines or mutants with autophagy mutants to check for genetic interactions. Otherwise, I could imaging affecting endocytosis would have an impact on nutrient starvation sensitivity.

We agree with the reviewer that the correlation between endocytosis and EPCS-dependent autophagy are at this point impossible to untangle.

Analyzing endocytic mutants will likely also not generate additional insight as these mutants are very likely to be hypersensitive to any stress and therefore not allowing any conclusion to be drawn.

Moreover, we believe that endocytosis at the EPCS is the first step into the autophagic pathway in plants which we identify here.

We have discussed this in more depth and will include the possibility that the observed increase in sensitivity can be linked to effects on endocytosis.

Minor comments:

- Line 30: are found associated with could be replaced by associates with

We have adapted the text.

- No need to tell this to these authors but shorter sentences are easier for readers to get their message. For example 1st sentence of the introduction. Also, it may be worth writing separate results and discussion sections. In the middle of the results section, they have done a recap of their results, which is unorthodox.

We respectfully disagree with the reviewer that it would be better to separate the results and discussion section. To answer this comment about the recap of the results, we have adapted the text to avoid recapping the results.

- Line 43: should be animals

adapted

- Line 45: nutritional should be nutrient

adapted

- Line 46: in a variety of different organisms could be in different organisms

adapted

- Line 61-70: this is not introduction, it belongs to the results section.

adapted

- Line 80: Results and discussion

adapted

- Line 95: references not formatted properly

adapted

- Line 99: CME has not been described before

adapted

- Figure 3K-EH domain seems to prefer PIP2 rather than PI3P. Don't see why the authors focus on Pi3P.

For simplicity, because another reviewer wanted us to expand these findings and also because the editor wanted us to tone down the conclusion based on the lipid strip results we have opted to remove the lipid binding data from the current manuscript. In the meantime, we have generated structural data on these domains and will include the lipid binding properties into an independent publication.

Reviewer #2 (Remarks to the Author):

This study attempts to link TPC components (AtEH1 and AtEH2) with the regulation of autophagosome formation at the ER-PM contact sites in Arabidopsis. Unfortunately the authors were unable to provide mechanistic role of AtEH1 and AtEH2 in the regulation of autophagy. Despite large amount of protein localization data, key experiments are missing, as described in the major comments.

Major comments

1. Fig. 2B. T-DNA inheritance experiments lack statistical analysis.

We respectfully disagree with the reviewer that statistical analysis is essential to support the conclusions drawn.

We have shown extensively in Van Damme et al., 2006 and Gadeyne et al., 2014 that mutants in TPC subunits (including mutants in *ateh2*) are male sterile and cannot transfer their T-DNA via the male in back-cross experiments to wild type.

Here we show by back-cross experiments that this is the same for the *ateh1* (+/-) mutant (0 out of 12) and that transfer can be established by introducing our complementing fusion constructs.

2. Fig 3B, 4D and 5H. Control cells show strong signal in cytoplasm and nucleus, contradicting expected localization of AtEH1/Pan1. It is not clear if fluorescence in the nuclei was included into FRET-FLIM quantification and if so, what is the biological relevance of this data. Are FRET-FLIM results statistically significant? The decrease in the fluorescence lifetime is around 5%. The charts shown with trimmed Y-axis can be misleading. What do error bars show?

We agree with the reviewer regarding the statistical analysis of the FRET-FLIM data.

We would also like to refer to the fifth comment of reviewer 1 where we included AtEH1/Pan1 T1 combined with RFP-ARP3 which shows no interaction as control for FRET-FLIM.

In this version, we have adapted the legends, added statistics and described the axes better for all FRET-FLIM data.

Regarding the FLIM analysis, we included all signals, including the nuclear signal, to avoid any bias. We have also clarified this in more detail in the materials and methods. We believe that the nuclear localization sometimes observed in *N. benthamiana* is a consequence of the over expression. We also see nuclear labelling in the test combinations (see image below, arrow) and even in the complemented mutant under stress conditions (Figure 4H). The cytoplasmic localization is always there as we are dealing with a cytoplasmic protein which is recruited to the autophagosomes. Similar cytoplasmic localization is found for most ATG genes as well (e.g. ATG8, ATG6 etc). The FRET-FLIM images are generated according to lifetime, but have low resolution in contrast to confocal images. Please also see the images below, where AtEH1/Pan1 labeled puncta are clearly demonstrated in CLSM, but appear blurry in the FLIM images.

The measured fluorescence lifetime of GFP-AtEH1/Pan1 depends on the association with the acceptor fluorophore (RFP), which is also over expressed, it does not depend on its localization.

The lifetime of our GFP fusions in the nucleus and the cytoplasm is similar (see image above where the color-coded signal in the nucleus does not differ from that in the cytoplasm), so it does not interfere with the overall measurement. We have performed the analysis on at least 15 individual images for every combination without specifically selecting for the image plane containing the nucleus and our lifetime measurements are very consistent. Our FLIM results, quantified using the whole image, clearly show differences in lifetime for the interactions we report, whereas the control experiment which we have now added (Figure S9G) clearly shows lack of interaction. We therefore believe that our approach to analyze the interactions is robust and sufficient.

3. Fig. 3E and F. Authors do not elaborate on the differences in the number of structures, localization of GFP-AtEH1/Pan1 and colocalization efficacy.

We would like to point out that Figure 3E is an immunostaining of a fixed Arabidopsis leaf using antibodies, while Figure 3F originates from live cell imaging in *N. benthamiana*. This is why the images look different and the number of autophagosomes differs.

We have adapted the text and the figure legend to make this clearer.

4. Lines 220-224. This is too stretching; It reads like ER-PM contacts are exclusive sites for AtEH-labeled autophagosome. In vivo live imaging analysis using TIRF microscopy is required to conclude EPCS origin of AtEH-labeled autophagosomes.

Our conclusion that AtEH/Pan1 autophagosomes originate from EPCS is based on the interaction with VAP27, which is located at the EPCS (Wang et al., *curr biol* 2014).

We believe that live cell TIRF time-lapse imaging showing formation of autophagosomes at EPCS goes beyond the scope of this paper.

We have rephrased the text to make it clearer that the pathway can start there, but that this does not mean that this is the exclusive starting point.

5. Authors should provide analysis of AtEH/Pan1 localizations in the ATG-deficient background and elaborate on the lack of endocytic impairment phenotypes in the atg knockout lines of *Arabidopsis thaliana*.

We have combined GFP-AtEH1/Pan1 with atg5 and atg7 mutant backgrounds. We have included data showing that in those mutant backgrounds, AtEH/Pan1 can still be recruited to punctae, which however are not delivered to the vacuole (Figure 3K and L).

We are not aware of anyone having addressed endocytic impairment in atg mutant plants (e.g. FM uptake assays, BFA body formation). We believe that the reviewer wants us to discuss the fact that atg mutants are developmentally less affected than the reported endocytic mutants (e.g. TPLATE amiRNA, OE of AUXILLIN). In our view however, endocytosis and autophagy should not lead to similar mutant phenotypes. Endocytic flux also includes internalization for recycling (e.g. transcytosis, polarity) and therefore inhibiting removal from the PM (endocytic mutant) will have a different effect than inhibiting degradation (atg mutant). We have adapted the text where we describe our model to include the difference between endocytosis leading to recycling and endocytosis leading to degradation.

6. Lines 267-270, Fig. 6A. Increased number of YFP-ATG8 foci and elevated expression of ATG genes do not tell whether autophagic flux is enhanced. The authors should run Western blot YFP cleavage assay in the presence/absence of Concanamycin A. An alternative could be NBR1 Western blot. The same problem with Figure S9.

We respectfully disagree with the reviewer as elevated levels of ATG genes and the number of autophagosomes have been used in a number of studies to test autophagy activity. It is also suggested in the 'Autophagy Guidelines' (Klionsky et al 2012; 2016).

As suggested by the reviewer, we performed *N. benthamiana* infiltrations with AtEH/Pan1-mCherry + Citrine-ATG8 and compared it with Citrine-ATG8 alone via WB to check citrine cleavage. We observed that basal levels of Citrine-ATG8 degradation are substantial when expressed alone. These experiments did not allow us to draw significant conclusions on the increased autophagic flux. We did not include this in the current manuscript but we have added the data here below. The figure shows WB after the single infiltration of citrine-ATG8a over time (2days, 3days and 7 days) compared to the combined expression of citrine-ATG8a together with EH1/Pan1-mCherry.

WB: antiGFP

We have also introduced mCherry-ATG8e into our inducible AtEH1/Pan1 lines and have provided images showing strong colocalization between AtEH1/Pan1 foci and ATG8e upon induction (Figure 6E). Also here, we could not generate direct evidence for increased flux in these lines, although the induced over expression of AtEH1/Pan1 does cause an increased accumulation of autophagosomes.

The figure below shows a typical example of our attempts/problems to address flux by WB. The figure shows two technical replicas (same samples loaded twice) of our inducible OE line of AtEH1/Pan1 where we followed the degradation of mCherry-ATG8e. Although the first panel clearly shows increased flux over time, this was less obvious in our second replica. Measuring flux by ATG8 degradation in our hands varies too much to allow drawing solid conclusions and we prefer not to include these data in the manuscript. The observed cleavage of FP-ATG8, both in *N. benthamiana* as in *Arabidopsis*, is in line with high basal ATG8 degradation in planta (Bassham, methods for analysis of autophagy in plants, Methods 2015). The ATG8 degradation in planta is very different from the situation in yeast, where hardly any ATG8 degradation is observed in the absence of stress (Kim et al., Elife 2016 DOI: 10.7554/eLife.12245 Figure 5).

However, to address the question about autophagic flux, we silenced AtEH1/Pan1, which correlated with strongly reduced numbers of ATG8-positive puncta under nutrient starvation conditions (Figure 6G). We believe that these novel results, together with hypersensitivity to nutrient starvation of the RNAi lines (Figure 6H and Figure S10D), address the question of the role of AtEH1/Pan1 in autophagic flux.

7. Lines 275-293. Increased accumulation of non-lipidated ATG8 protein and AtEH-GFP foci does not indicate enhanced autophagic activity. What about ATG8 lipidation? Is autophagic flux also increased?

We would like to point out that ATG8 lipidation in plants is difficult to use as a readout of enhanced autophagy (Bassham, methods for analysis of autophagy in plants, Methods 2015). With respect to the question about autophagic flux, we would like to refer to our novel experiments described in the answer to the previous comment.

8. Fig. 6B-D. What does chart on panel B show? Statistical analysis is missing. There is no difference in the starvation response between WT and RNAi. ATG knockout mutants should be used as control. Autophagic flux should be measured instead of ATG8 gene expression, which is not directly indicative of autophagic activity.

We have adapted the axes, stating clearly that we measure normalized root length, made the text bigger, did statistical analysis and elaborated on the statistical method used in the legend.

We also performed novel experiments with carbon starvation where we compared our RNAi lines with the atg5 mutant. We observed similar hypersensitivity of the RNAi lines as atg5. We included these results in Figure S10D and commented in the text on the difference between -N and -C stresses.

9. Lines 305-313. Again, these experiments do not tell about autophagy level.

We agree that we did not address autophagy directly in these experiments. However, as we also state in the text, these results show that EPCS residents are linked to stresses normally involving induction of autophagy. A recent paper in PNAS also links SYT1 recruitment to the EPCS under high salt and low nutrient conditions, resulting in expansion of the EPCS (Lee et al., PNAS 2019). We have adapted the text to make it clear that these experiments link EPCS residents to stress responses involving autophagy.

10. Fig. S11A. Statistical treatment is missing.

We have included appropriate statistical analysis.

11. Generally it is not clear from the presented data, whether proposed upregulation of autophagy in AtEH/Pan1 overexpressors and susceptibility of AtEH/Pan1 KD to starvation is caused by pleiotropic effect on endocytosis or by autophagy-specific response.

We agree with the reviewer that this remains to be determined. Our current result demonstrates that the number of autophagosomes is linked directly to the expression level of AtEH1/Pan1; AtEH1/Pan1 localized to autophagosomes and its RNAi mutant are more susceptible to autophagy stresses. Taking these results together, we have made a careful conclusion that AtEH1/Pan1 regulates autophagy. However, as we are dealing here with proteins which are likely to be involved in both processes, it is not trivial to separate both functions. Moreover, it also remains to be determined whether an endocytic block might not activate increased autophagy.

As also stated above in the answers to the questions from reviewer 1, we decided not to test endocytosis mutants (e.g. ap2m) for their susceptibility to starvation as this would likely not allow the generation of any significant conclusions.

Minor comments

1. Fig. 1. VCA-like domain is not shown.

VCA-like domain is represented by “A” in the figure. We have exchanged “VCA” in the text by “the acidic motif”.

2. Fig 2A. Pollen viability is typically assessed using Alexander staining. SEM is a suboptimal approach as shrinkage might be caused by exposure to vacuum.

We performed alexander staining in our first TPLATE manuscript in 2006 (Van Damme et al., 2006). In this publication we show that the shrunken pollen is, at least at some point, still viable according to alexander staining. Our earlier data also shows a clear correlation between the genetic (heterozygous) and the morphological phenotype (shrunken pollen). We have identified 1:1 ratios between normal and shrunken pollen for several independent mutants in TPC subunits, while this phenotype was absent in WT pollen (Van Damme et al., 2006; Gadeyne et al., 2014). We therefore believe that it is unlikely that our observations of shrunken pollen are caused by the vacuum.

3. Immunofluorescent images lack negative controls

We have included representative immunofluorescent images of anti AtEH/Pan1 together with GFP-HDEL in Figure S6E as negative control for the immunofluorescent images.

4. Fig. 3B. Scale bars are missing

We have added scale bars to all FRET FLIM images.

5. Fig 4A lacks the input and has a weird negative actin-like band for AtEH1/Pan1 sample

The image was taken by an illuminator that has a light from above, so some reflection unexpectedly was seen in that region (caused by some air bubbles, see below).

In the current manuscript, we have expanded the part of the blot which is shown (now including the actin alone as well) as well as the gel in Figure 4.

6. Lines 182-184. This conclusion is far-fetched. Have the authors indeed shown that AtEH proteins are required for autophagosome formation? Rephrase

We believe that our novel data showing reduced amounts of ATG8-positive foci upon silencing AtEH1/Pan1 show that the latter plays an important role in autophagosome formation. We have adapted the text to avoid any over-interpretation.

7. Lines 192-193. How can vacuolar co-localization shown in Fig. 3L suggest functional role of TPLATE in autophagy? Rephrase

We agree with the reviewer that we did not provide evidence for a functional role for the TPLATE subunit in autophagy. We have rephrased the text to: the vacuolar co-localization of TPLATE with AtEH/Pan1 suggests that AtEH/Pan1 proteins likely function in autophagy together with other TPC subunits.

8. Lines 210-212. Perplexing, rephrase. Reference(s) is missing

We have rephrased the text and added a reference.

9. Lines 212-214. Premature conclusion. The authors have not shown yet that AtEH proteins are required for autophagosome formation.

We have adapted the text and would also like to refer here to our answer to comment number 6.

10. Fig 5D, there are no autophagosomes on the micrograph

We have provided a better representative image for ATG4.

11. Fig 5G contradicts the statement that VAP27-1 is localized at EPCS.

For simplicity, and also because another reviewer had an issue with this panel, we have removed this panel from the revised manuscript.

12. Fig. 3D, 5E and F. GFP-AtEH1/Pan1 shows 100% co-localization with RFP-ATG8e and VAP27-1-YFP positive puncta, while only a few RFP-ATG8e co-localize with VAP27-1-YFP puncta.

We believe this is because AtEH/Pan1 over expression induces autophagy.

In Figure 5E, we do not over express AtEH/Pan1 and therefore we do not induce autophagy leading to low levels of colocalization.

13. Sections starting on lines 215 and 228 can be combined.

We have combined the sections.

14. Line 229. "AtEH1 regulated autophagy". When readers have reached this point they do not know yet that AtEH1 regulates autophagy.

We have adapted the text.

15. Lines 249-251. This conclusion would be much stronger if also/instead shown for GFP-ATG8 labeled autophagosomes.

We would like to point out that we have combined VAP27-1 with AtEH/Pan1 and ATG8 in Figure S6B. We have rephrased the text to refer to the triple colocalization with ATG8.

16. Please revise the abbreviations used in the text, some lack explanation, e.g. CME.

We have adapted the text.

Reviewer #3 (Remarks to the Author):

This paper describes a comprehensive set of experiments implicating the Arabidopsis proteins, AtEH1 and AtEH2, in actin-mediated autophagy. The authors show that these EH domain-containing proteins, which have previously been implicated in endocytosis as part of the TPLATE complex, link the process of autophagy with endocytosis and the actin cytoskeleton. A diverse set of experiments were conducted to support the authors conclusions including AtEH1 and AtEH2 downregulation, cell biological studies to demonstrate autophagosome localization of functional AtEH1/2-fluorescent protein fusions as well as colocalization of AtEH1 and AtEH2 with markers to various components of the TPLATE complex and ARP2/3, in vitro actin binding, and immunogold electron microscopy. Overall, this is a nice and thorough study. The manuscript is well written and provides a timely follow up on previous work by the authors implicating the actin nucleation machinery in mediating autophagy in plants. In previous work, they showed that a component of ARP2/3, NAP1, is involved in actin-mediated autophagy. Their discovery that AtEH1 and AtEH2 are part of this actin-modulated machinery is novel and exciting. However, I list some concerns below that the authors might want to address:

1). The paper contains a significant amount of data showing colocalization between AtEH1/2-GFPs and various markers such as autophagosomes, TPLATE components and ARP3. Looking at the image data alone, it is clear that AtEH1/2 colocalizes with these various components. That said, there

appears to be a glaring lack of quantitative data to support their colocalization work. Would a Pearson's correlation analysis be a useful quantitative tool for this? I ask for some sort of quantitative data to support the colocalization images because in looking at the image data alone, it appears that in some cases, colocalization is almost 100% (e.g. GFP-ATEH1 and RFP-ATG8e in Figure 3D). Is it always the case that colocalization is almost 1:1 between AtEH-GFP and other red-emitting RFP markers? I would have raised a concern about bleedthrough but I note from the methods section that the authors take this into account by simultaneous scanning. Nonetheless, some sort of quantitation of

colocalization would improve the paper and allow more transparency as it would provide information on how many cells were actually sampled, how many independent infiltration/imaging experiments were conducted etc. Such information should be stated in the figure legends.

We agree with the reviewer that the colocalization is sometimes striking. We took special care to avoid bleed through by sequential imaging and the choice of which protein was tagged with which fluorophore. An example here are the colocalization analyses in *N. benthamiana* between the AtEH/Pan1 proteins and the other endocytic players where we always tagged the AtEH/Pan1 with mCherry to ensure that all signal in the green channel was coming from the interacting GFP-fusion. There are also plenty of observations in the manuscript lacking this full colocalization. For example, Figure S1D (FreeGFP and AtEH2/Pan1-mCherry), Figure S1E-F (AtEH/Pan1-mCherry and NBR1-GFP), Figure S4F (AtEH2/Pan1 with AP1/2B1-GFP) and Figure S8F (GFP-AtEH1 and VAP27-1-YFP deltaMSD).

We believe that adding Pearson's correlations here is not optimal as the density of the punctae makes it often impossible to prevent image saturation at these points and saturation is not advisable in combination with correlation analysis.

We obviously base our observations on several independent experiments and have also stated this more clearly in the figure legends.

2). In relation to point 1 above, the statistics used to support data in which authors have done some sort of quantification, need clarification. For example, the FLIM dataset is nice. However, I think the authors missed indicating what type of statistics was done on the FLIM work. I assume "test" in the parenthesis in the figure legend is Student t-test? What are p values? Put asterisks on the FLIM bar graphs to indicate significance. Same point I made above, the authors should indicate in the FLIM analysis, how many cells were sampled. I know they state 15 cells in the methods section but to be more transparent, this information should be indicated in the figure legend.

We have adapted the statistics, put asterisks and stated the number of cells analyzed more clearly in the figure legends.

3). As a follow up to my two earlier concerns, the authors might want to check carefully for any missing information in their figure legends for data (and supplemental figures) in which quantification/statistical analysis is involved. Another example are results on low nutrient sensitivity of AtEH/PAN1 knockdowns (Figure 6B). What is the Y axis for Figure 6B. Root length? It seems like the RNAi lines depicted in the bar graphs were not subjected to statistical analysis. Such analysis is needed to show that knockdowns are indeed more sensitive than wild type to nutrient (low N and low C) stress.

We have adapted axes legends, added statistics and increased the font.

4). The authors show that AtEH1 and AtEH2 proteins associate with foci through the N-terminal EH domains. Based on lipid overlay assays, the authors show that such association might be due to binding of AtEHs to anionic phospholipids, most notably PI3P. This dataset could be strengthened by another set of controls or if not too difficult verified by other phospholipid binding assays that is more robust than lipid overlays. With regard to the additional controls, the lipid overlays shown in Figure 3K only have lipid blots for the His-tagged EH domains (i.e. HIS-EH1.1 and HIS-EH1.2). No negative controls are provided. It might be important to include HIS-CC-A (i.e. HIS c-terminus) or HIS alone as negative controls. I raise this issue because these lipid overlay blots can be notorious for non-specific binding. The negative controls would assure readers that the binding patterns shown in Figure 3K are genuinely related to the EH domains. Better yet, the authors might want to consider liposome assays as an alternative to lipid overlay blots.

We agree with the reviewer that lipid strip data is optimally supported by liposome assays. We decided to remove the lipid strip data in this version of the manuscript. We did this for simplicity, because reviewer number 1 also had a remark on these findings and because the editor wanted us to tune down the conclusion based on the lipid strip results. In the meantime, we have generated structural data on these domains and prefer to include the lipid binding properties into an independent publication.

5). Is the RFP-ARP3 fusion used by the authors to demonstrate colocalization with AtEH1/2-GFP and autophagosome marker functional (Figure 4C)? I think this would be useful for the readers to know. arp3 mutants have a distinct trichome phenotypes so expression of the RFP-ARP3 in the arp3 mutant should complement the trichome defects if it is functional. I might just be missing a reference or prior work by the authors demonstrating functionality of the ARP3-RFP thus this concern could be addressed by simply adding the appropriate reference. I ask about functionality of the ARP3-RFP because it would potentially be a useful tool for many readers interested in plant actin.

Arabidopsis ARP3 has been used before to complement the yeast arp3 mutant, but this was done with a different fusion than the one we generated here. We are mainly using the ARP3 here to add to the interaction of AtEH with actin and to link AtEH to Pan1. We believe that the clear recruitment observed, in contrast to the cytoplasmic localization observed in the absence of AtEH/Pan1 over expression, argues against the ARP3-RFP fusion being dysfunctional. We believe that performing experiments to prove the functionality of our ARP3 fusion by complementing the trichome developmental defects of the arp2/3 mutants lies beyond the scope of this paper and would unnecessarily delay the publication of this work.

6). The authors demonstrate that overexpressing AtEH/PAN1 promotes autophagy. 24h postinduction, they show that AtEH/PAN1 is at the plasma membrane in the root meristem and are more prominent in autophagosome foci in the elongation zone. I have reservations about the latter claim (Figure 6F). The images of root cells shown with foci clearly have emerging root hairs. Thus, this region of the root can no longer be called the elongation zone because root hairs typically emerge in the differentiation/maturation zone. If the authors want to claim that AtEH/PAN1 is more prominent in the elongation zone compared to the meristem, they need to show cells in the elongation zone that have not yet reached full maturity. I think a few microns or mm from the regions shown in Figure 4E should have cells in the “true” elongation zone. The authors can then claim that AtEH1/PAN1 are in foci in both the root elongation and differentiation zone.

We agree with the reviewer and we have adapted the text stating that the foci are most prominent in the differentiation/maturation zone.

7). The references in 15 and 16 (Coutts et al) is a duplicate

We have adapted the references.

Reviewer #4 (Remarks to the Author):

This study provides direct evidence showing that AtEH1 and AtEH2 are able to interact with F-actin and ER-PM contact site resident protein VAP27-1, and regulate the formation of autophagosome. Since both AtEH1 and AtEH2 were members of the TPLATE complex (TPC) which regulates plant endocytosis, another new finding in this manuscript is that AtEH1 protein could recruit both TPC and ARP2/3 protein to the autophagosome, indicating the alternative mechanism for autophagosome formation at plant EPCS. In plant cell, the knowledge of how actin participates in autophagosome formation is still very limited. So the current study is potentially of great interest to the field. The experiments are in general well designed and performed, but there were a few major concerns to be addressed for the authors:

1. The authors reported AtEH1 protein drive autophagosome formation at the ER-PM contact site. For this point: immuno-gold TEM of AtEH1 data was provided, but it is difficult to specify the EPCS localization of AtEH1 protein from the single images in Figure 5A, the corresponding statistic analysis should be involved to confirm the AtEH1's specific localization at EPCS or at least similar distribution pattern as VAP27-1. Also the localization of VAP27-1 is different when co-expressed with ATG8 (Fig 5E) compared to when co-expressed with AtEH1 (Fig 5F) in *N. Benthamiana*, is this due to the specific interaction between AtEH1 and VAP27-1 to mediate autophagosome formation? To figure out this point: the co-localization relation between AtEH1, ATG8 and VAP27-1, especially in *Arabidopsis* should be analysed.

We do not want to claim specific localization of AtEH1/Pan1 at EPCS. AtEH/Pan1 is homogenously distributed on the PM, which is what we show in our complemented mutant lines in the roots and the spinning disc imaging (Figure 2). What we claim here is that AtEH/Pan1 can be recruited to EPCS by VAP27-1, next to its function as part of the TPLATE complex in general endocytosis at the PM. We have unpublished data that AtEH/Pan1 proteins in our complemented mutant lines under normal conditions are at all times colocalizing with the other TPLATE complex subunits at the PM.

When AtEH/Pan1 and VAP27-1 are co-over expressed however, AtEH/Pan1 is dragged to VAP27-1 and we believe that the altered localization is merely a proof of their tight interaction. This is clear from the current Figure S7 where combining AtEH1/Pan1 with VAP27-1 alters the mobility of the AtEH/Pan1 autophagosomes. Therefore, the differences in localization of VAP27 and AtEH1/Pan1 (e.g. Figure 5E and 5F) are likely due to its interaction of the over-expressed proteins.

We have performed triple co-localization in *N. benthamiana* (Figure S6B). As expected, the overlap between AtEH/Pan1 and ATG8 is very high and AtEH/Pan1 gets strongly recruited to the VAP27-positive EPCS, resulting in a prominent triple co-localization.

In the previous manuscript, we combined AtEH/Pan1 with VAP27-1 in *Arabidopsis* (current Figure S6C). However, previously we did not yet include colocalization data between induced AtEH/Pan1 punctae and ATG8. Now, we generated new data showing colocalization between AtEH/Pan1 foci and ATG8 in *Arabidopsis* and included the results in Figure 6E. We believe that combining this on top with VAP27-1 goes beyond the scope of this manuscript, given the *Arabidopsis* data we already have and the fact that we already provide triple colocalization data in *N. benthamiana*.

Combining AtEH and VAP27 antibodies for IEM is unfortunately not possible as both antibodies were raised in mice.

2. The authors showed the evidence that AtEH1 co-localized with both ATG8 and ATG6, further the co-localization with ATG6 indicated AtEH1's function in early stage of autophagosome formation, also in the Fig 7B, the AtEH1 were labelled in the PAS. Please provide the direct evidence to confirm this result, for example the time course dynamic distribution pattern of both ATG8 and AtEH1.

In an attempt to address this comment experimentally, we performed a pressure induced experiment, similar to NAP1. For NAP1, pressure-induced autophagosome formation resulted in ATG8 accumulation (Wang et al., curr biol 2016).

Unfortunately, we did not succeed. ATG6+ATG8+AtEH/Pan1 triple expressing cells turned out to be very difficult to find, and in those cells where we could confirm expression of all constructs, already formed autophagosomes made assessing the timing of recruitment impossible.

We agree with the reviewer that the evidence for AtEH/Pan1 presence in the PAS is currently only supported by our ATG6 colocalization and results from GFP-AtEH1/Pan1 (atg5/7 mutant). We have adapted the legend of the model presented in Figure 7B to make this clear.

3. Different truncated construct were made and confirmed that the EH domain is essential for the lipid binding (Fig 3K), whether the construct with EH domain alone could label punctate structures and PM associated foci as AtEH1-T1 which contain two EH domain? In the end of the study, it is showed the AtEH1-T1 poorly co-localized with ATG8, could authors clarify what the punctate structure is and whether this structure related to the autophagosome formation?

We believe the reviewer asks about the nature of the foci observed with the T1 construct, which contains both EH domains, yet do not recruit ATG8. The localization of the T1 construct highly resembles PM foci which we observe with other TPC subunits fused with mRUBY3 in *N. benthamiana* (see image below which shows TWD40-1-mRUBY3 foci).

Those foci are much smaller than the AtEH/Pan1-labelled autophagosomes and show fast recruitment and dissociation from the PM.

We therefore think that those are endocytic foci and have included this hypothesis in the text. For simplicity, we prefer not to work this out in detail. Also, because we removed the lipid binding data of the EH domains, based on the comments of reviewer 1, 3 and the editor, we also moved the localization of the truncations to supplemental material (Figure S9).

4. To figure out AtEH1's important role in autophagosome formation, the RNAi lines were constructed and some autophagosome phenotypic analysis were made. Would be very important and interesting to check in these RNAi lines, how ATG8-RFP affected, and also whether the autophagosome formation, VAP27-1 and ARP protein localization will be affected?

We thank the reviewer for pointing this out. It is indeed a remaining question. However, combining our RNAi lines with several markers and assessing the role of reduced AtEH/Pan1 expression on autophagosome formation, EPCS morphology and ARP localization seems to us beyond the scope of this initial manuscript where we identify the existence of this pathway. We did however partially addressed this comment as we performed anti ATG8 immunofluorescence in our RNAi lines with and without nutrient stress. Silencing AtEH1/Pan1 correlated with reduced numbers of ATG8-positive puncta under nutrient starved conditions (Figure 6G).

In addition, there are some minor concerns are listed:

1. Line 95: Sanchez-Rodriguez et al., Mol Plant 2018, not cited in the proper way, also not involved in the reference list;

We have adapted this.

2. In the reference list: #15 and #16 are exactly the same.

We have adapted this.

REVIEWERS' COMMENTS:

Reviewer #1 (Remarks to the Author):

The authors addressed my comments. below are a few text edits they should modify.

- Line 38-40. Sentence needs reference(s).
- Line 40-41- the statement that autophagosomes are formed by membrane invagination is simply wrong. They are de novo formed vesicles.
- Line 180-181: the terminology is wrongly used here. Autophagosomes do contain ATG8, the authors should use phagophore or isolation membrane.

Reviewer #2 (Remarks to the Author):

The authors have addressed most of my comments in the revised manuscript.

Reviewer #4 (Remarks to the Author):

In the updated manuscripts by Wang et al, the authors re-organized the text or figures, for example: authors adapted the legend of the model to clarify AtEH1's function in autophagosome formation according to their new experimental data, also the lipid binding data was removed from the manuscripts. For the 1st question, authors added some data, which made the interaction between AtEH and VAP27-1 more convincing, so I would think most of the questions were properly answered. Only for the 4th question, authors offered additional immunofluorescence to show that ATG8e indeed was affected in the RNAi lines, which is good. However, the immunofluorescence under non-stress condition should be involved to check whether the RNAi lines with altered autophagosome numbers or not; also, the "y axis" in figure 6G is "number of autophagosomes" per what, per cell? Authors need to make it clear.

In addition, the figure 5G and 5H showed in the text as 5H-I, which need to be corrected.

REVIEWERS' COMMENTS:

We thank all reviewers again for their contributions and critical advice for us to improve this manuscript.

Reviewer #1 (Remarks to the Author):

The authors addressed my comments. below are a few text edits they should modify.

- Line 38-40. Sentence needs reference(s).

> Done.

- Line 40-41- the statement that autophagosomes are formed by membrane invagination is simply wrong. They are de novo formed vesicles.

> We have modified it to "(autophagosomes) formed through vesicle transport and membrane expansion"

- Line 180-181: the terminology is wrongly used here. Autophagosomes do contain ATG8, the authors should use phagophore or isolation membrane.

> We have changed it to 'phagophore'.

Reviewer #2 (Remarks to the Author):

The authors have addressed most of my comments in the revised manuscript.

Reviewer #4 (Remarks to the Author):

In the updated manuscripts by Wang et al, the authors re-organized the text or figures, for example: authors adapted the legend of the model to clarify AtEH1's

function in autophagosome formation according to their new experimental data, also the lipid binding data was removed from the manuscripts. For the 1st question, authors added some data, which made the interaction between AtEH and VAP27-1 more convincing, so I would think most of the questions were properly answered. Only for the 4th question, authors offered additional immunofluorescence to show that ATG8e indeed was affected in the RNAi lines, which is good. However, the immunofluorescence under non-stress condition should be involved to check whether the RNAi lines with altered autophagosome numbers or not;

> In the revised manuscript, we have provided images of MDC labelled autophagosomes at non-stress and stressed conditions as a confirmation of our immune-ATG8 data. Clearly, not many autophagosomes (using MDC positive foci as proxy) are observed at non-stress condition for both Col-0 or RNAi lines, whereas, more autophagosomes appear in Col-0 at stress condition. We hope that we answered the concern of the reviewer sufficiently with these new data.

also, the “y axis” in figure 6G is “number of autophagosomes” per what, per cell? Authors need to make it clear.

>We have clarified the axis in the legend.

In addition, the figure 5G and 5H showed in the text as 5H-I, which need to be corrected.

>We have corrected this.